

# Radiative feedbacks of dust-in-snow over East Asia in CAM4-BAM

Xiaoning Xie[1], Xiaodong Liu[1,2], Huizheng Che[3], Xiaoxun Xie[1], Xinzhou Li[1], Zhengguo Shi[1],
Hongli Wang[4], Tianliang Zhao[5], and Yangang Liu[6]

[1]SKLLQG, Institute of Earth Environment, Chinese Academy of Sciences, Xi'an 710061, China
[2]University of Chinese Academy of Sciences, Beijing 100049, China
[3]Key Laboratory for Atmospheric Chemistry, Institute of Atmospheric Composition, Chinese Academy of Meteorological Sciences, CMA, Beijing 100081, China
[4]Shaanxi Radio and TV University, Xi'an 710119, China
[5]Key Laboratory for Aerosol-Cloud-Precipitation of China Meteorological Administration, Nanjing University of Science Information &Technology, Nanjing 210044, China
[6]Environmental and Climate Sciences Department, Brookhaven National Laboratory, Upton, NY 11973-5000, USA

*Correspondence to:* Xiaoning Xie (xnxie@ieecas.cn)

**Abstract.** Dust-in-snow on the Tibetan Plateau (TP) could reduce the visible snow albedo by changing surface optical properties and remove the snow cover by increasing snowmelt, which leads to a significant positive radiative forcing and remarkably alters the regional energy balance and the East Asian climate system. This study extends our previous investigation in dust-radiation interactions to investigate the dust-in-snow radiative forcing (SRF) and its feedbacks on the regional climate and the

5  dust cycle over East Asia by use of the Community Atmosphere Model version 4 with a Bulk Aerosol Model parameterization of the dust size distribution (CAM4-BAM). Our results show that SRF increases the East Asian dust emissions significantly, by 13.7% in the spring, in contrast to −7.6% of decreased dust emissions by the dust direct radiative forcing (DRF). SRF also remarkably enhances the whole dust cycle, including dust transports, dry and wet depositions over East Asia. The simulated results show that the combined effect of DRF and SRF increases the dust emissions by 5.1%, and enhances the overall dust

10  cycle over this region. Further analysis reveals that these results are mainly due to the regional climatic feedbacks induced by SRF over East Asia. By reducing the snow albedo over the TP, the dust-in-snow mainly warms the TP to enhance its thermal effects by increasing the surface sensible and latent heat flux, and then increases the aridity and westerly winds over Northwest China, in turn enhances the East Asian dust cycle. Additionally, the dust-in-snow also accelerates snow melting, reduces the snow cover, and then expands the East Asian dust source region area, which results in increasing the regional dust emissions.

15  Hence, a significant feature of SRF on the TP can create a positive feedback loop to enhance the dust cycle over East Asia.





## 1 Introduction

A large amount of desert dusts from East Asia arid and semi-arid regions are emitted into the atmosphere, which can be carried over the wide downwind regions including the eastern China and the Pacific Ocean, and also deposited in snow over the Tibetan Plateau (Wake et al., 1994; Zhang et al., 1997; Zhao et al., 2006). The dusts can significantly affect global and regional energy

balance, climate, and hydrological cycle by dust direct radiative forcing and dust-in-snow radiative forcing (Ramanathan et al., 2001; Shao et al., 2011; Mahowald et al., 2014; Huang et al., 2014; Qian et al., 2015).

Dusts in the atmosphere can directly absorb and scatter the thermal (longwave) and solar (shortwave) radiation labeled by dust direct radiative forcing (DRF). Note that the importance of DRF in general circulation models (GCMs) has been recognized for many years (Tegen and Lacis, 1996; Miller and Tegen, 1998; Yue et al., 2009; Mahowald et al., 2014). On the

global scale, the dust DRF is about $-0.4$ W m$^{-2}$ with a range between $-0.30$ and $-0.6$ W m$^{-2}$ estimated by the current models described by Huneeus et al. (2011), as reviewed by Kok et al. (2017). Furthermore, Kok et al. (2017) claimed that the DRF estimated by these current GCMs is underestimated due to inaccurate emitted dust size distribution, and the new size distribution results in a larger DRF as $-0.2$ W m$^{-2}$ from $-0.48$ to $+0.20$ W m$^{-2}$. The regional DRF over Asia is significantly higher than that at the global scale due to the larger dust loading, influencing the regional climate (Lau et al., 2006; Zhang et

al. 2009; Han et al. 2012; Sun et al., 2012; Guo and Yin et al., 2015; Gu et al., 2016). A so-called Elevated Heat Pump effect due to atmospheric heating by elevated absorbing aerosols strengthens large-scale atmospheric circulation and enhances the precipitation in the late boreal spring and early summer season over the foothills of the Himalayas and northern India (Lau et al., 2006). The aerosol-heating reduces the Tibetan and Himalayan snowpack cover with $6-10\%$, neglecting the greenhouse warming (Lau et al., 2010). The net surface and TOA radiative fluxes are decreased by DRF and cause a surface cooling over

East Asia and increase the regional local stability (Zhang et al., 2009). Dust loading in spring and summer alerts the East Asian summer monsoon through affecting atmospheric circulation and thermal structures induced by DRF (Sun et al., 2012; Guo and Yin, 2015). Gu et al. (2016) claimed that the circulation and precipitation responses to DRF are different over South/East Asia and North Africa, which is depending on the relative location of dusts to the rainfall band.

Depositions of light absorbing aerosols on snow (e.g., black carbon and dust) can reduce the visible snow albedo by changing

surface optical properties and remove snow cover by increasing snowmelt in entirely or partially snow covered areas, resulting in a significant positive radiative forcing (Hansen and Nazarenko, 2004; Xu et al., 2009; Huang et al., 2011; Qian et al., 2015). Based on previous studies (Hansen and Nazarenko, 2004; Hansen et al., 2005), IPCC (2007) showed the radiative forcing range of $0.10\pm0.10$ W m$^{-2}$ induced by aerosol-in-snow at the global scale, whereas IPCC (2013) adopted a radiative forcing of $+0.04$ ($+0.02$ to $+0.09$) W m$^{-2}$ according to results of Bond et al. (2013). The Tibetan Plateau (TP) is a vast elevated plateau with an

average elevation of 4, 500 meters, located in Asia (Figure 1). In addition to being close to the Taklamakan (one of the largest sand deserts in the world) and Gobi deserts, the TP also has several deserts (e.g., Qaidam Basin desert) within it and is also near the industrial regions in Indian subcontinent and eastern China. Hence, there exists a larger amount of deposition on snow of black carbon and dust aerosols over the TP due to the high industrial and natural emissions in Asia (Xu et al., 2009; Ming et al., 2013; Qu et al., 2014). Over this region, the particles of dusts are the dominant insoluble impurities compared with black





carbon in terms of particle mass (Ming et al. 2013; Qu et al., 2014), and they further claimed that the impact of dust aerosols on snow albedo and dust-induced surface radiative forcing exceed those of black carbon over the TP, mainly because of larger dust loading. The aerosol-induced snow albedo perturbation generates much larger positive surface radiative flux changes with $5-25$ W m$^{-2}$, during spring over the TP (Flanner et al., 2009; Qian et al., 2011). Furthermore, Qian et al. (2011) claimed that

absorbing aerosol in snow can warming the TP around $1.0°$C, which can influence the East Asian and South monsoon through the TP's thermal and dynamical forcing.

Dust cycles, including dust emissions, transports, as well as dry and wet depositions, are altered by DRF through affecting the atmospheric vertical thermal structures and surface wind speed. The mechanism of PBL (the planetary boundary layer) (Miller et al., 2004; Perez et al., 2006; Heinold et al., 2007) was firstly proposed by to explain the reduction of dust emissions

induced by DRF (Perlwitz et al., 2001). It was described that the surface negative net DRF by dusts reduces the turbulent flux of surface sensible heat and reduces PBL mixing. A positive feedback between DRF and dust emissions are shown through the PBL mechanism over North Africa due to the surface positive net DRF. An alternative mechanism was proposed that DRF produces an anomaly in the surface pressure, especially on the edge of the dust layer, resulting in affecting the circulation and wind speed (Ahn et al.,2007; Heinold et al., 2008).

In our previous study (Xie et al., 2018), we have shown using the improved Community Atmosphere Model version 4 with a Bulk Aerosol Model parameterization of the dust size distribution (CAM4-BAM) that DRF decreases the East Asian dust cycle owing to the negative surface radiative forcing through the PBL mechanism, which is opposite to the enhancement of North African dust emissions induced by DRF. Considering that the dust cycle change by the dust-in-snow radiative forcing (SRF) over East Asia has not studied previously, here we extended the the DRF effects (Xie et al., 2018) to systematically to

investigate the SRF and the regional climate change and the dust cycle change induced by the SRF over East Asia, using the identical CAM4-BAM model, and also compared with the DRF effects discussed in Xie et al. (2018). The rest of this paper is structured as follows. In Section 2, we first review the improved CAM4-BAM model based on Albani et al. (2014) and Xie et al. (2018), and describe the experiment design, and furthermore the model performance is also evaluated against the temporal and spatial observations about the snow cover and the surface temperature. The model results about dust radiative forcing and

its radiative feedbacks are discussed for the area of study in Section 3. Further discussions and conclusions are summarized in Sections 4 and 5, respectively.

## 2    Model evaluation and experiment design

### 2.1    CAM4-BAM and experiments

The CAM4 model detailedly described by Neale et al. (2010) is the atmospheric component of the Community Climate System

Model version 4 (CCSM4). The CAM4-BAM model considers a subbin fixed size distribution of externally mixed sulfate, sea salt, organic carbon, black carbon, and dust by Tie et al. (2005). The improved CAM4-BAM model was proposed based on three major aspects including the optimized soil erodibility maps with respect to each of the macroareas, updated optical properties of dusts with realistic absorption parameters, and a new size distribution for dust emissions, which has a better representation



of the dust cycle, most notably for the improved size distribution (Albani et al., 2014). This improved model can be used to investigate East Asia dust cycle and display its DRF over this region (Xie et al., 2018). Black carbon and mineral dust in snow were represented in the Snow, Ice, and Aerosol Radiative (SNICAR) component (Flanner et al., 2007; 2009), which has been used to investigate the aerosol-in-snow forcing at the global scale (Flanner et al., 2009) and at the regional scale (Qian et al.,

5  2011).

In this work, the improved CAM4-BAM model adopts the finite volume (FV) scheme for the dynamical core with a higher horizontal resolution ($0.9° \times 1.25°$), and with 26 levels in the vertical direction. All the model simulations were run for the year 2000 and constrained with the observed sea surface temperature, sea ice concentrations, and atmospheric forcings including solar irradiance, tropospheric and stratospheric ozone, and greenhouse gases during this period. We conducted three numerical

experiments covering 21 years with a one year spin up, one with both DRF and SRF (Case1), one with the DRF and without the SRF (Case2), the other one without the DRF and the SRF (Case3), as summarized in Table 1. It is noted that, here we only consider the dust aerosols in these three numerical experiments, and neglect the radiative properties of other aerosols including sulfate, sea salt, organic carbon and black carbon in the improved CAM4-BAM model. Based on these three experiments including Case 1, Case 2, and Case 3, we can derive the DRF (Case2−Case3), the SRF (Case1−Case2), and the total radiative

forcing (DRF+SRF, Case1−Case3). Hence, we use the results of the differences between these three experiments to investigate the climatic feedback of SRF and the dust cycle changes induced by SRF over East Asia, also as compared with the results in terms of DRF.

## 2.2   Model evaluation

This subsection assesses the climatological features including the simulated dust aerosol optical depth at 550 nm (AOD), dust

deposition, snow cover and surface temperature in order to evaluate the impacts of dust-in-snow forcing over the TP. Figure 2a shows the monthly dust AOD over the TP from the CAM4-BAM model. It shows that the dust AOD has the largest values (> 0.06) in MAM (March-April-May), likely because of the higher frequency of dust storms over this region in MAM. Figure 2b displays the spatial distribution of the simulated MAM dust AOD from CAM4-BAM. This simulated spatial distribution shows that the dust AOD has larger values over these two dust source regions including Gobi and Taklamakan deserts, which is much

larger than 0.2, especially over the Taklamakan desert. Figure 2c shows the monthly dust deposition over the TP including dry, wet and total (dry+wet) depositions. The result also indicates the largest dust deposition in MAM. This phenomenon of the monthly variation of dust deposition is very similar with the dust AOD (Figure 2a). It is noted that the dry deposition of dusts is much larger than the wet deposition probably because of less rain over Northwest China. Figure 2d shows that the total dust deposition exhibits two peaks over the two dust source regions. Additionally, there exists larger dust deposition over the

western and northeastern parts of the TP, which will lead to a larger SRF, because these regions have several deserts, and are near the Taklamakan desert and the Gobi desert.

Figure 3 shows the monthly mean snow cover fraction (SCF) and surface temperature over the TP from the model and observations (MODIS SCF and CRU surface temperature data). The MODIS data shows a larger SCF in the winter and spring over the TP, with the maximum reaching approximately 25% in January (Figure 3a). The corresponding minimum SCF





occurs in July and August. Overall, the CAM4-BAM model can capture the monthly variations of SRF, with the minimum in summer and maximum in winter and spring. But, the model overestimates the amplitude of the monthly variations of SCF, with overestimated SCF in winter and underestimated SCF in summer. The overestimated SCF variation is due probably to the model's coarse horizonal resolution ($0.9° \times 1.5°$) and smooth terrain (Qian et al., 2011; Lee et al., 2013). Figure 4a shows that the MAM persistent snow covered areas with SRF > 50% are located in the western TP (including Pamir Plateau, Tianshan, and Kunlun mountains), the southeast TP (Hengduan mountain), Himalayas mountain, and Qilian mountain. The model can capture these most persistent snow covered areas (Figure 4b). In contrast to the case of SCF, the data from the Climatic Research Unit (CRU) shows a lower surface temperature in winter and spring, and a higher surface temperature in summer (Figure 3b). Furthermore, the CAM4-BAM model can better capture the spatial distribution (Figure 4c) and the monthly variations of the surface temperature (Figure 4d), although it also overestimates the amplitude of monthly variations in the surface temperature.

## 3 Dust radiative forcing and its radiative feedbacks

### 3.1 Dust cycle changes induced by SRF

It has been well recognized that mineral dusts can influence the atmospheric vertical thermal structures and surface wind speed by DRF, which affects the dust cycle through various mechanisms (Perlwitz et al., 2001; Miller et al., 2004; Perez et al., 2006; Heinold et al., 2007; Ahn et al., 2007; Heinold et al., 2008; Colarco et al., 2014; Xie et al., 2018). However, the dust cycle change induced by the SRF over East Asia has not been systematically studied previously, despite reported large surface positive SRF over the TP (Flanner et al., 2009; Qian et al., 2011). This subsection fills this gap to examine the SRF-induced dust cycle changes in the MAM over East Asia, including dust emissions, dust transport (which is the vertically integrated dust flux), and the dry and wet depositions. Comparison with the DRF is also made to estimate their relative contributions.

Figure 5 shows the spatial distribution of the dust cycle changes induced by DRF (left column), SRF (middle column), and the total radiative forcing (DRF+SRF, right colume) over East Asia in MAM; the corresponding averaged values are summarized in Table 2 over the East Asian dust source area ($75°E-115°E$ and $25°N-50°N$). Figures 5a shows that DRF decreases the dust emissions over this region by $-8.8$ Tg season$^{-1}$, with a magnitude of $-7.6\%$ in Table 2. The corresponding dust transport, dry and wet depositions in Figures 5d, 5g and 5j are also decreased over this region by DRF, with the magnitudes of $-6.4\%$, $-5.7\%$, and $-1.8\%$, respectively. Figure 5b shows that SRF remarkedly enhances the dust emissions over the East Asian dust source area, with 14.78 Tg season$^{-1}$ (13.7%), which is statistically significant. It is noted that the changes of the dust emissions induced by SRF are approximately 2 times larger than that by DRF. The changes of dust transport, dry and wet depositions (Figures 5e, 5h, and 5k) are also significantly enhanced by SRF with 6.9%, 11.9%, and 4.7%, respectively.

Figures 5c, 5f, 5i and 5l show the changes in dust cycle induced by the dust total radiative forcing. The dust emissions is significantly enhanced (in Figure 5c) by the dust total radiative forcing over East Asia with an increase of 5.98 Tg season$^{-1}$ by 5.1%. That is because the changes of dust emissions induced by the SRF are much larger than that by the DRF. The dry and wet depositions are also increased (5.5% and 2.9%) in Figure 5i and 5l by the dust total radiative forcing. Figure 5f shows the dust transport is enhanced over the northern region of the East Asian dust source region although the averaged value of the



dust transport is slightly decreased by −0.9% (shown in Table 2) over this region due to the decreased dust transport over the southern region. In summary, the dust total radiative forcing enhances the dust cycles, because the SRF-induced enhancement is much stronger than the diminishment caused by DRF.

## 3.2 Dust radiative forcing and the dust-induced changes in surface properties

It is noted that these dust cycle changes induced by the DRF, the SRF and the dust total radiative forcing (Figure 5) are mainly due to the corresponding radiative forcing and its climatic feedbacks. This section examines the dust-induced radiative forcing including DRF and SRF and its climatic feedbacks, especially about the SRF.

Figure 6a shows the spatial distribution of the simulated surface albedo in MAM for Case 1. Evidently, there exists a larger broadband surface albedo over the TP, especially over the western TP due to larger SCF (Figure 4b). Dust-in-snow can
decrease the snow albedo over entirely or partially snow covered areas as mentioned above. Figure 6b shows the dust-in-snow significantly reduces the broadband surface albedo over the TP and its surrounding mountains, especially over the western TP (reaching over −0.1). The decrease in snow albedo mainly results from a positive feedback process: absorbing aerosols deposited on snow → reducing surface albedo → increasing surface net solar radiation → increasing surface temperature → reducing snow fraction and depth → finally reducing surface albedo, which was proposed by Qian et al. (2011).

Figure 7 shows that the dust-induced changes in the surface radiative forcing and surface temperature in MAM by the DRF, the SRF and the total radiative forcing. The DRF displays a surface negative radiative forcing over East Asia, where it is much larger and statistically significant over the Taklamakan and Gobi deserts (Figure 7a). That is mainly because that the higher AOD over these two dust source regions (Figure 2b) scatters and absorbs the solar radiation and exerts a larger surface negative radiative forcing. This surface negative radiative forcing reduces the surface sensible heat, and then decreases vertical mixing
within the PBL and the wind speed at the surface, in turn decrease the regional dust emissions (labeled by the PBL mechanism) over East Asia (Xie et al., 2018). It is noted that this PBL mechanism was firstly shown and proved by Miller et al. (2004), Perez et al. (2006) and Heinold et al. (2007). Figure 7b shows the surface temperature is also decreased slightly between −1°C to 0°C over East Asia, likely because of the surface negative radiative forcing.

In contrast, Figure 7c shows that SRF displays a significant surface positive radiative forcing over the whole TP and the
surrounding mountains, especially over the western TP (reach above 20 W m$^{-2}$). The decreased surface albedo over the TP (Figure 6b) causes increasing surface net solar radiation and shows a surface positive radiative forcing over this region. This positive surface radiative forcing significantly warms the whole TP, especially the western TP (beyond 2°C) in Figure 7d. Figure 7e shows the dust total radiative forcing exerts a significant and larger surface positive radiative forcing over the whole TP, whereas a significant and smaller surface negative radiative forcing over the Taklamakan and the Gobi deserts. Over the TP,
the SRF mainly determines the total radiative forcing and the DRF determines the total radiative forcing over the deserts and the wide downwind regions. Hence, the surface temperature significantly increases over the TP due to the larger total positive forcing in Figure 7f, which is determined by the SRF. Hence, we will focus on the changes in the surface properties induced by the SRF in the followings.



Figure 8a shows the spatial distribution of the changes in the SCF induced by the SRF. The SCF is significantly decreased by the SRF over the whole TP and its surrounding mountains, where the maximum decrease of the SCF can reach above $-15\%$. The warming TP (Figure 7d) due to dust-in-snow accelerates snow melting, reduces the snow cover, and then expands the dust source region area, resulting in enhancing the regional dust emissions. Figure 8b shows the significant increase in the surface latent heat flux (LHF) by the SRF. That is due to the increased soil moisture induced by the enhanced snowmelt over the TP. Additionally, the increased surface precipitation during spring and summer by the SRF also increases the soil moisture in the following subsection. The warming TP also increases the regional surface sensible heat flux (SHF) in Figure 8c. The surface total heat flux (LHF+LHF) shows a larger value over the TP, especially over the western TP (reaching over 10 W m$^{-2}$). Hence, dust-in-snow over the TP can warm the TP and enhance its thermal effects by increasing the surface LHF and SHF, resulting in affecting the Asian climate.

## 3.3 Dust-induced climatic feedbacks

It is known that the TP as the Third Pole can influence the Indian and East Asian summer monsoonal and inland precipitation through its dynamical and thermal effects (e.g., Boos and Kuang, 2010; Wu et al., 2012; Liu et al., 2013; Shi et al., 2014; Sha et al., 2015). Recently, Qian et al. (2011) claimed that the absorbing aerosols (especially the black carbon) in snow over the TP affects the East and South Asian monsoon climate and hydrological cycle by the enhanced TP thermal effects. Here, we concentrate on the enhanced TP thermal effects due to dust-in-snow on the climate over the arid and semiarid regions of Northwest China.

Figure 9 show the changes in the zonal wind component and the Omega in a vertical cross section at $75°E-115°E$ in MAM induced by the SRF. It shows the anonymous westerly wind over the north of the TP (mainly including Northwest China) and the anonymous easterly wind forced by the SRF over the south of the TP in Figure 9a, which are generally statistical significant over these two regions. That is because that the enhanced TP thermal effect due to increasing the surface LHF and SHF induced by the SRF increases the south-north temperature gradient, resulting in a westerly wind anomaly over the north of the TP (Schiemann et al., 2009; Li and Liu, 2015). Additionally, we show the spatial distribution of the mid-level westerly winds (Figure 10a) and the surface wind speed (Figure 10b) over East Asia, indicating that the the mid-level westerly winds and the surface wind speed are also significantly increased over Northwest China. The mid-level westerly winds have been recognized as one of the major factors for the long-distance dust transport process to the North Pacific Ocean and to North America and beyond from East Asian dust sources (Wilkening et al., 2000; Guo et al., 2017), while the surface wind speed remarkably affects dust emission rate by influencing the dust saltation process (Shao et al., 2011). Hence, the increased westerly wind enhances the dust emissions and dust transport, in turn increases the whole dust cycle. Additionally, the enhanced TP thermal effect induced by SRF enhances the upward vertical velocity over the TP and the downward vertical velocity over Northwest China (or intensifies subsidence) in MAM from the vertical distribution in Figure 9b, which is also shown in Figure 10c based on the spatial distribution. This result leads to the decreased surface precipitation over Northwest China and the enhanced surface precipitation over the TP in Figure 10d. Note that the enhanced TP thermal effect by SRF lasts from spring to summer, which has been shown in Figure 11. It shows the downward vertical velocity is enhanced (decreased) in summer





(Figure 11a), resulting in the significant decreased (increased) surface precipitation (Figure 11b) over the north of the TP (over the TP). Hence, the SRF significantly decreases the surface precipitation in spring and summer and then enhances the regional aridity in Northwest China, resulting in enhancing the regional dust emissions. In generally, the dust-in-snow mainly warms the TP and then increases the aridity and westerly winds by the enhanced TP thermal effects, in turn enhances the East Asian

dust cycle.

## 4 Further discussions

Figure 12 shows the changes in the dust emissions by the SRF in other seasons including JJA, SON, and DJF compared to MAM. It shows that the dust emissions are all enhanced by the SRF in all the seasons (MAM: 14.78 Tg season$^{-1}$, JJA: 0.78 Tg season$^{-1}$, SON: 1.56 Tg season$^{-1}$, and DJF: 5.86 Tg season$^{-1}$). The enhancement of dust emissions in MAM induced by

the SRF is much larger than that in other seasons, which can account for 64% of the annual total increase in dust emissions. That is mainly because that largest dust deposition on snow is in MAM over the TP, which exerts a significant radiative forcing, climatic feedbacks, and changes in dust emissions in this season. Therefore, it is quite reasonable to focus on the dust radiative forcing and its feedbacks on climate and dust cycle in MAM in our work.

The above results suggests the main reasons with respect to the enhanced dust cycles over East Asia by SRF, which is

illustrated in the schematic diagram (shown in Figure 13). Dust aerosols are emitted from East Asian dust sources with arid and less rain and deposited on snow over the TP. The dust-in-snow over the TP reduces the surface albedo (Figure 6b) and warms the TP (Figure 7d), and then increases the westerly winds (Figure 9a) and the surface wind speeds (Figure 9b) through enhancing the south-north temperature gradient, and enhances the aridity over Northwest China (Figures 10d and 11b) by intensified subsidence. These enhanced aridity and stronger westerly winds can increase the dust emissions, and enhance the

whole dust cycle (Figures 5d, 5e, 5h, and 5k). Additionally, the dust-in-snow also accelerates snow melting, reduces snow cover (Figure 8a), and then expands the dust source region area, resulting in increasing the dust emissions. Hence, a significant feature of SRF over the TP can create a positive feedback loop to enhance the dust emissions (as summarized in Figure 13).

Similar to our previous study (Xie et al., 2018), we compares the changes in the East Asian dust cycle by the dust radiative forcing with the North African dust emissions. Table 3 shows that the dust emissions are significantly enhanced with 8.9% by

the DRF over North Africa, through both the strengthened large-scale circulation and the PBL mechanism (Xie et al., 2018). The changes in dust emissions by the SRF are much smaller than that by the DRF over North Africa, mainly due to less snow cover and negligible SRF over this region (figures not shown). Therefore, the systematic comparative analysis reveals that the change in dust cycle over North Africa is dominated by DRF (Table 3) whereas it is controlled by SRF over East Asia due to the existence of the TP, as shown in Table 2.



## 5    Concluding Remarks

A large amount of desert dusts from East Asia arid and semi-arid regions are deposited on snow over the Tibetan Plateau (TP), the dust-in-snow reduces the visible snow albedo by changing surface optical properties and removes the snow cover by increasing snowmelt, leading to a significant positive radiative forcing (labeled by SRF). SRF over the TP can influence the

regional climate and dust cycle over East Asia through enhancing the TP thermal effects.

In this study, the improved CAM4-BAM model was used to investigate the SRF and its feedbacks on the climate system and the dust cycle over East Asia. The CAM4-BAM model's results show that the SRF increases the dust emissions significantly by 14.78 Tg season$^{-1}$ with the magnitude of 13.7% in the spring due to the existence of the TP, and then remarkably enhance the whole dust cycle including dust transports, dry and wet depositions over East Asia. Compared to the decreased dust emissions

by $-8.80$ Tg season$^{-1}$ with $-7.6$% (through the PBL mechanism) induced by the dust direct radiative forcing (DRF), the increased effects on dust emissions by SRF are much more significant. Simulation results show that the total effects of DRF and SRF can increase the dust emissions by 5.98 Tg season$^{-1}$ with 5.1%. By reducing snow albedo over Tibetan Plateau, the dust-in-snow mainly warms the TP and enhances the TP thermal effects by increasing the surface sensible and latent heat flux, and then increases the aridity and westerly winds over Northwest China, in turn enhances the regional dust cycle. Additionally,

the dust-in-snow also accelerates snow melting, reduces snow cover, and expands the dust source region area, resulting in increasing dust emissions. In generally, a significant feature of SRF over the TP can create a positive feedback loop to enhance the dust cycle, which is overall summarized in Figure 13.

It is noted that black carbon (BC) deposited on snow over the TP mainly from South Asia and East Asia (Xu et al., 2009; Wang et al., 2015) also displays a significant positive forcing over this region (Flanner et al., 2009; Qian et al., 2011). Here,

we only consider the radiative forcing of the dust-in-snow over the TP ignoring the radiative forcing of the BC-in-snow in our study. Hence, we maybe underestimate the total radiative forcing of the absorbing aerosols over the TP. Additionally, we focus on the East Asian arid and semi-arid regions in order to investigate the dust cycle changes induced by SRF in this paper. Qian et al. (2011) pointed out that the radiative forcing of the absorbing aerosols can substantially influence the South and East Asian monsoon climate and the regional hydrological cycle through the TP thermal effects using the CAM3.1 model.

It is noted that the atmospheric dust burden and deposition rate of dusts were much higher (approximately from 2 to 4 times) during the Last Glacial Maximum (LGM) compared to current climate mainly due to increased winds speeds, and the decrease in intensity of the hydrological cycle, as well as the expansion of dust source areas (Mahowald et al., 2006; Maher et al., 2010; Albani et al., 2012). The increased dust-in-snow over the TP due to higher atmospheric dust loadings may show a much larger positive radiative forcing and more significantly create the positive feedback loop to enhance the dust cycle in LGM. Hence,

we will investigate the SRF and its feedbacks on the East Asian climate and the dust cycle during the LGM in the future.

*Acknowledgements.* This work was jointly supported by National Key Research and Development Program of China (2016YFA0601904) and the National Natural Science Foundation of China (41690115, 41572150). Z. Shi is supported by CAS "Light of West China" Program. Y.





Liu is supported by the US Department of Energy's Atmospheric System Research (ASR) program. The data of CRU temperature is acquired from http://www.cru.uea.ac.uk/data and the data of MODIS snow cover is from https://nsidc.org/data/modis/data_summaries/#snow.



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



**Table 1.** Description of the Model experiments in this work. Here DRF represents dust direct radiative forcing and SRF is defined as dust-in-snow radiative forcing.

| Experiments | Simulated time | DRF | SRF |
|---|---|---|---|
| Case1 | 21 years (1 years spinup) | Yes | Yes |
| Case2 | 21 years (1 years spinup) | Yes | No |
| Case3 | 21 years (1 years spinup) | No | No |





**Table 2.** The March-April-May (MAM) averaged dust emissions (Tg season$^{-1}$) , transport (g m$^{-1}$ s$^{-1}$), dry deposition (Tg season$^{-1}$), and wet deposition (Tg season$^{-1}$) over the East Asian dust source area (75°E−115°E and 25°N−50°N) in Case1, Case2, and Case3, as well as their corresponding differences between these three experiments.

|  | Dust emission | Dust transport | Dry deposition | Wet deposition |
|---|---|---|---|---|
| Case1 | 122.40 | 1.08 | 68.92 | 36.99 |
| Case2 | 107.62 | 1.01 | 61.59 | 35.33 |
| Case3 | 116.42 | 1.09 | 65.33 | 35.96 |
| DRF (Case2−Case3) | −8.80 (−7.6%) | −0.07 (−6.4%) | −3.74 (−5.7%) | −0.63 (−1.8%) |
| SRF (Case1−Case2) | 14.78 (13.7%) | 0.07 (6.9%) | 7.33 (11.9%) | 1.66 (4.7%) |
| DRF+SRF (Case1−Case3) | 5.98 (5.1%) | −0.01 (−0.9%) | 3.59 (5.5%) | 1.03 (2.9%) |





**Table 3.** The March-April-May (MAM) averaged dust emissions (Tg season$^{-1}$), transport (g m$^{-1}$ s$^{-1}$), dry deposition (Tg season$^{-1}$), and wet deposition (Tg season$^{-1}$) over the North African dust source area ($-20°$E$-35°$E and $10°$N$-30°$N) in Case1, Case2, and Case3, as well as their corresponding differences between these three experiments.

|  | Dust emission | Dust transport | Dry deposition | Wet deposition |
|---|---|---|---|---|
| Case1 | 276.21 | 2.70 | 177.28 | 16.21 |
| Case2 | 292.21 | 2.84 | 187.23 | 16.51 |
| Case3 | 268.34 | 2.79 | 168.54 | 14.73 |
| DRF (Case2−Case3) | 23.87 (8.9%) | 0.05 (1.8%) | 18.69 (11.1%) | 1.78 (12.1%) |
| SRF (Case1−Case2) | −16.00 (−5.5%) | −0.14(−4.9%) | −9.95 (−5.3%) | −0.30 (1.8%) |
| DRF+SRF (Case1−Case3) | 7.87 (2.93%) | −0.09 (−3.2%) | 8.74 (5.2%) | 1.48 (10.0%) |





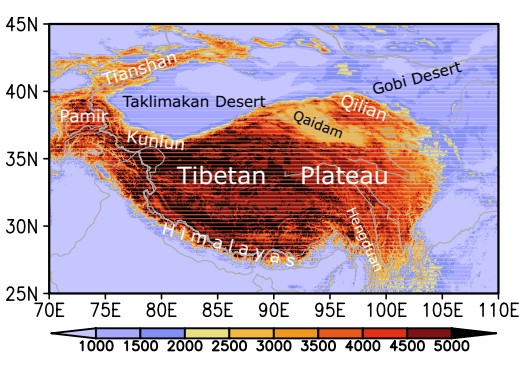

**Figure 1.** Terrain of the Tibetan Plateau (unit: m) including major mountains (Kunlun, Himalayas, Hengduan, Qilian and Tianshan Mountains), Pamir Plateau, Qaidam Basin deserts and its surrounding deserts (Taklimakan and Gobi deserts).



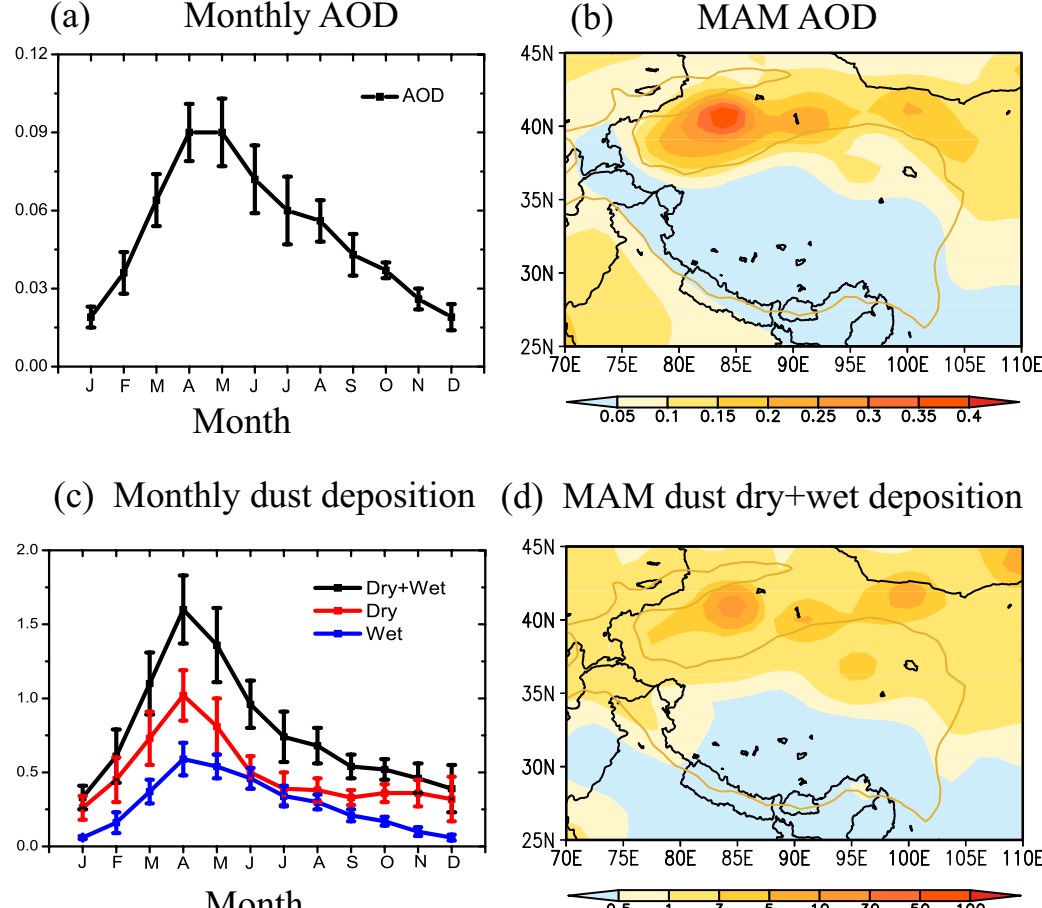

**Figure 2.** (a) Monthly dust AOD and spatial distribution of the March-April-May (MAM) averaged AOD from the CAM4-BAM model over the Tibetan Plateau (70°E−110°E and 25°N−45°N); (c) Monthly dust deposition ($\mu$g m$^{-2}$ s$^{-1}$) including dust dry, wet and dry+wet deposition and (d) spatial distribution of the MAM averaged total dust deposition (dry+wet, $\mu$g m$^{-2}$ s$^{-1}$) over this region. Note that the error bars (a, c) represent the standard deviation of the corresponding variables, and the yellow-contour area (b, d) indicates the plateau above 2500 m.




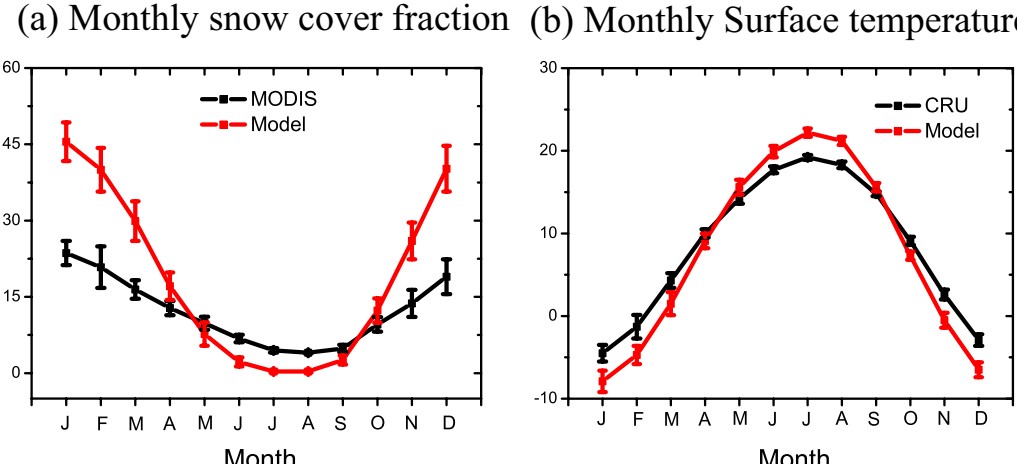

**Figure 3.** (a) Monthly snow cover fraction (%) from MODIS and Model; (b) monthly surface temperature (°C) from CRU and Model over the Tibetan Plateau (25N−45N, 70E−110E), where the error bars represent the standard deviation of the corresponding variables.




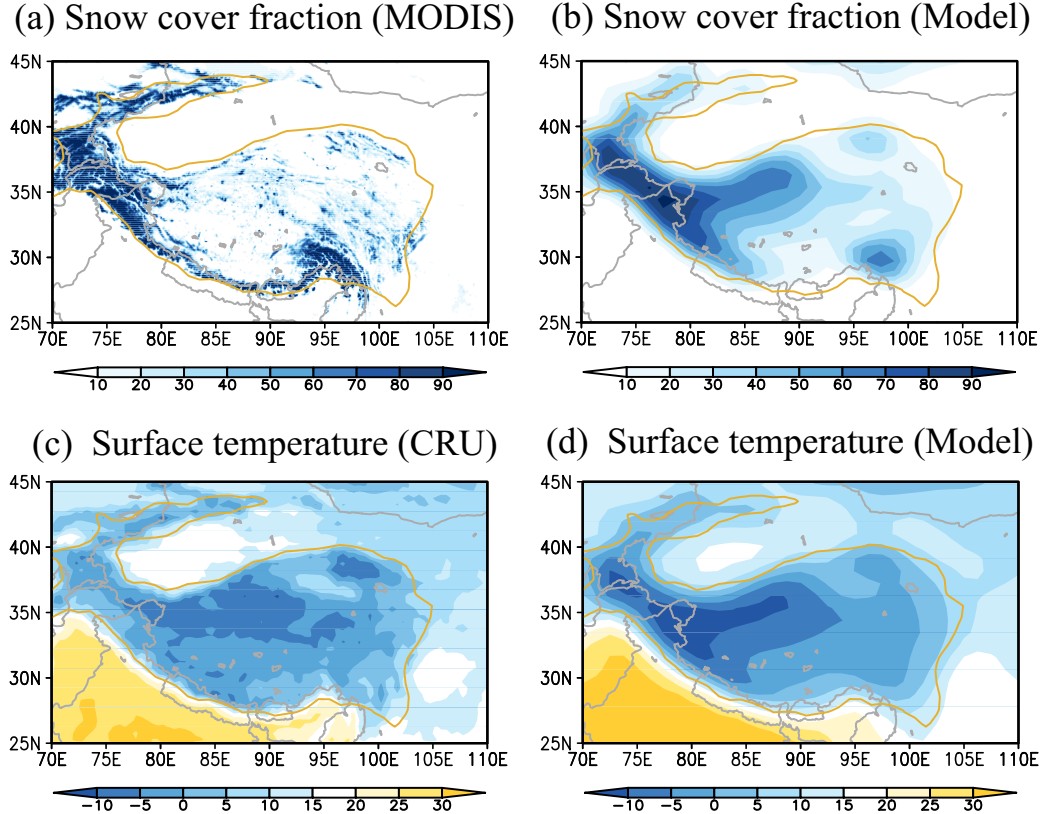

**Figure 4.** Spatial distribution of the MAM averaged (a) snow cover fraction (%) derived from MODIS for 2002−2012 and (c) surface temperature (°C) from CRU for 1979−2012, compared with the simulated MAM averaged (b) snow cover and (d) surface temperature for 20 model years over Tibetan Plateau. The yellow-contour area indicates the plateau above 2500 m.



**Figure 5.** Dust cycle changes including (a, b, and c) dust emissions (g m$^{-2}$ season$^{-1}$), (d, e, and f) dust transport (g m$^{-1}$ s$^{-1}$), (g, h, and i) dust dry deposition (g m$^{-2}$ season$^{-1}$), and (j, k, and l) dust wet deposition (g m$^{-2}$ season$^{-1}$) in MAM induced by dust direct radiative forcing (DRF), dust-in-snow radiative forcing (SRF), and total forcing (DRF+SRF). The oblique line represents the grid points where the changes pass the two-tailed t-test at the 5% significance level.



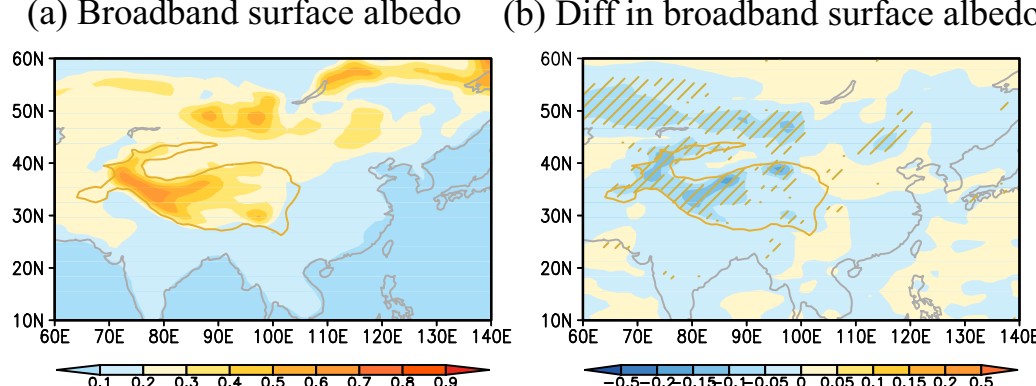

**Figure 6.** (a) Spatial distribution of the broadband surface albedo in MAM (Case 1) and (b) its changes between Case 1 and Case2, which are induced by dust-in-snow radiative forcing (SRF). The oblique line represents the grid points where the changes pass the two-tailed t-test at the 5% significance level. The yellow-contour area indicates the plateau above 2500 m.





**Figure 7.** Spatial distribution of the changes in (a, c, e) the surface radiative forcing (W m$^{-2}$), (b, d, f) the surface temperature (°C) in MAM induced by dust direct radiative forcing (DRF), dust-in-snow radiative forcing (SRF), and total radiative forcing (DRF+SRF). The oblique line represents the grid points where the changes pass the two-tailed t-test at the 5% significance level. The yellow-contour area indicates the plateau above 2500 m.





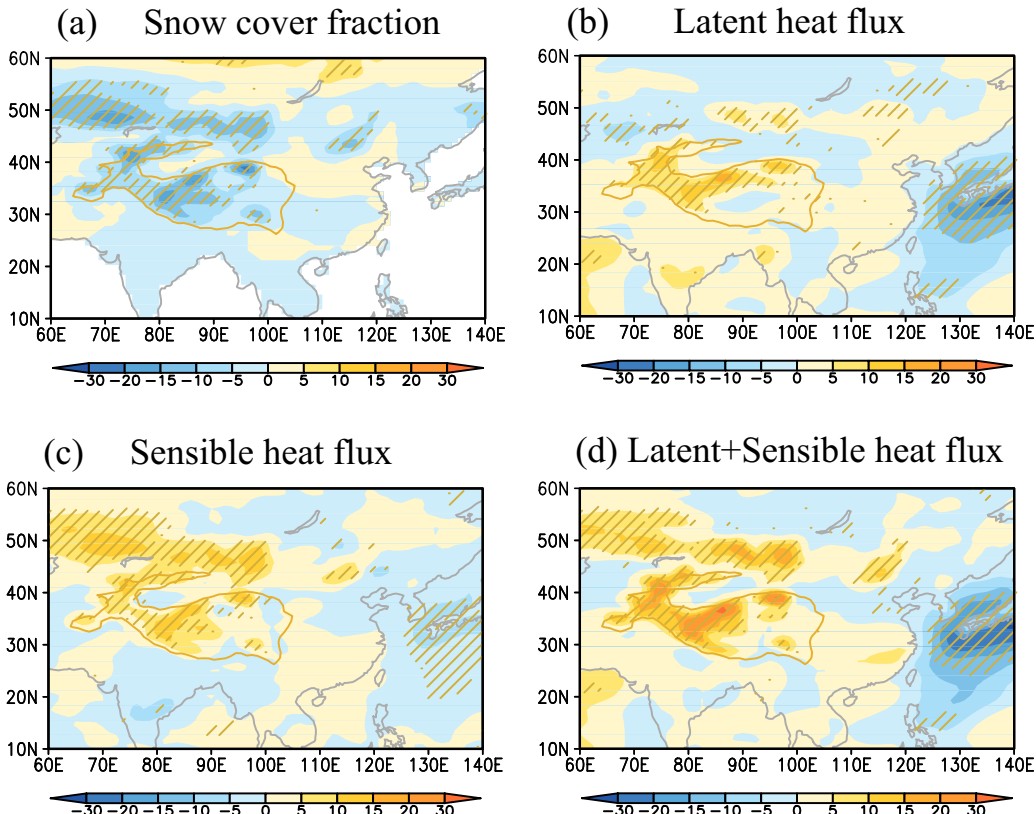

**Figure 8.** Spatial distribution of the changes in (a) the snow cover fraction (%), (b) the surface latent heat flux (W m$^{-2}$), (c) the surface sensible heat flux (W m$^{-2}$), and (d) the surface latent+sensible heat flux (W m$^{-2}$) in MAM induced by the dust-in-snow radiative forcing. Here the oblique line or the grey shaded area represent the grid points where the changes pass the two-tailed t-test at the 5% significance level. The yellow-contour area indicates the plateau above 2500 m.





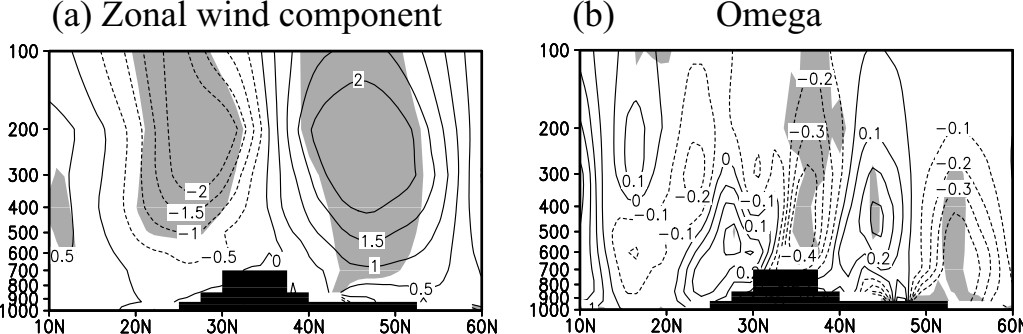

**Figure 9.** Changes in (a) the zonal wind component (m s$^{-1}$) and (b) the Omega ($0.01 \times$ Pa s$^{-1}$) in a vertical cross section at 75°E−115°E in MAM induced by the dust-in-snow radiative forcing. Here the grey shaded area represents the grid points where the changes pass the two-tailed t-test at the 5% significance level. The black shaded area indicates the plateau topography.





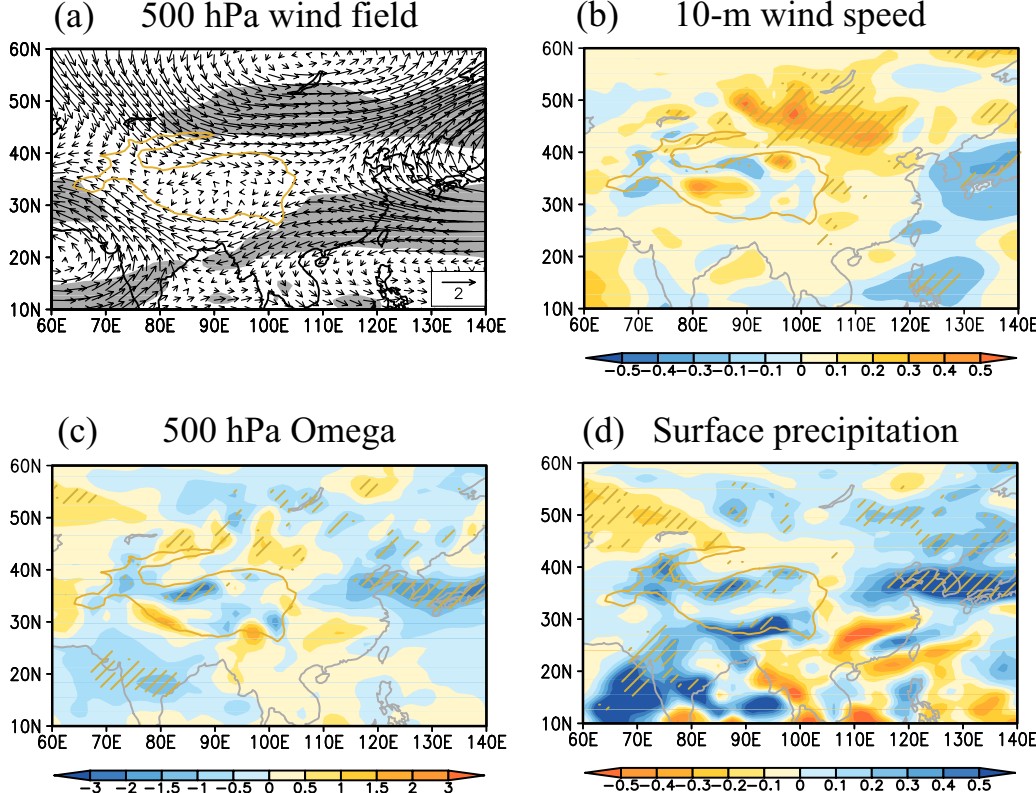

**Figure 10.** Spatial distribution of the changes in (a) the 500 hPa wind field (m s$^{-1}$), (b) the 10-m wind speed (m s$^{-1}$), (c) the 500 hPa Omega (0.01 × Pa s$^{-1}$), and (d) the surface precipitation (mm day$^{-1}$) in MAM induced by the dust-in-snow radiative forcing. Here the oblique line or the grey shaded area represent the grid points where the changes pass the two-tailed t-test at the 5% significance level. The yellow-contour area indicates the plateau above 2500 m.




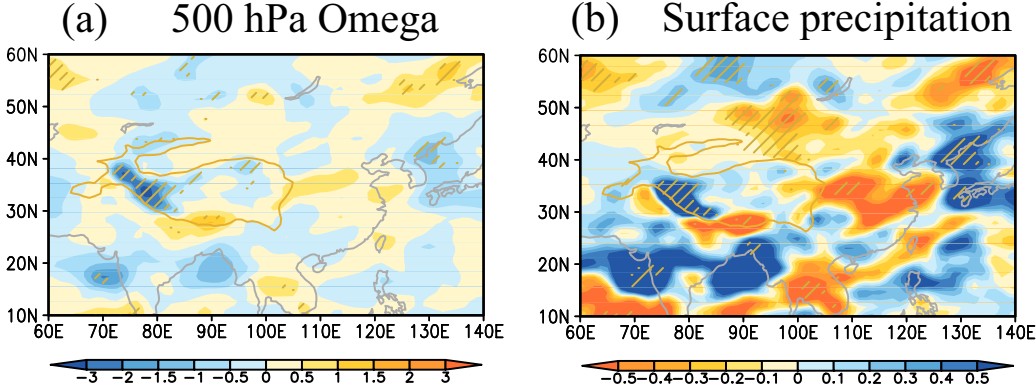

**Figure 11.** Spatial distribution of the changes in (a) the 500 hPa Omega ($0.01 \times$ Pa s$^{-1}$) and (b) the surface precipitation (mm day$^{-1}$) in June-July-August (JJA) induced by the dust-in-snow radiative forcing. The oblique line represents the grid points where the changes pass the two-tailed t-test at the 5% significance level. The yellow-contour area indicates the plateau above 2500 m.





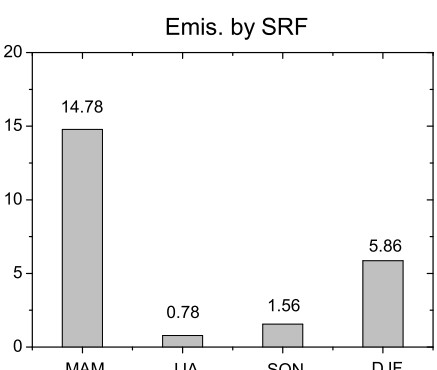

**Figure 12.** Seasonal changes in dust emissions (Tg season$^{-1}$) induced by the dust-in-snow radiative forcing (SRF) including March-April-May (MAM), June-July-August (JJA), September-October-November (SON), and December-January-February (DJF).



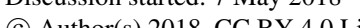

Figure 13. A schematic depiction of the feedback mechanism between dust emission and dust-in-snow radiative forcing.