# Peer review of "Radiative feedbacks of dust-in-snow over East Asia in CAM4-BAM"

_Atmospheric Chemistry and Physics, 2018_

## Referee Comment (RC1) · Anonymous Referee #1 · 17 Jun 2018

Review Comments for "Radiative feedbacks of dust-in-snow over East Asia in CAM4-BAM" by Xie et al.

The authors systematically investigated the responses of dust emissions, transport, and deposition to dust direct and in-snow radiative effects over East Asia. This work could help to improve the understanding of dust radiative effects and feedbacks in this region. The manuscript is generally well written and particularly I like Figure 13 which concisely summarizes the possible dust-in-snow radiative feedback. I have a few comments for improving the manuscript. Although most of my comments are minor, they need to be addressed properly before the manuscript can be considered for publication.

**1.** Page 2, Lines 11-13: As the authors mentioned, Kok et al. (2017) showed that inaccurate dust size distribution could lead to nontrivial biases in modeled DRF. Is it accurate enough by using the Bulk Aerosol Model (BAM) scheme embedded in CAM to represent dust size distributions as done in the present study?

**2.** Page 2, Lines 24-34: A number of recent references on advancing the understanding of BC/dust-in-snow effects are missing here. For example, several studies (e.g., Flanner et al., 2012; Liou et al., 2014; Dang et al., 2016; He et al., 2017b, 2018a) have shown the significant impacts of snow grain shape (spherical vs. nonspherical) and aerosol-snow mixing state (internal vs. external) on BC/dust-snow albedo forcing. Further studies also investigated the effects of snow grain packing (e.g., He et al., 2017a) and aerosol size distribution in snow (e.g., Schwarz et al., 2013; He et al., 2018b) on aerosol-snow interactions. Since the aerosol-in-snow effect is the focus of this study, I suggest including these recent references here. In addition, in terms of BC/dust deposition over the TP (Lines 32-34), some latest observational studies (e.g., Lee et al., 2017; Li et al., 2018; Zhang et al., 2018) can also be included here.
References:
Dang, C., Q. Fu, and S. Warren: Effect of Snow Grain Shape on Snow Albedo, J. Atmos. Sci., 73, 3573–3583, doi: 10.1175/JAS-D-15-0276.1, 2016.
Flanner, M. G., X. Liu, C. Zhou, J. E. Penner, and C. Jiao: Enhanced solar energy absorption by internally-mixed black carbon in snow grains, Atmos. Chem. Phys., 12(10), 4699–4721, doi:10.5194/acp-12-4699-2012, 2012.
He, C., Y. Takano, and K.-N. Liou: Close packing effects on clean and dirty snow albedo and associated climatic implications, Geophys. Res. Lett., 44, doi:10.1002/2017GL072916, 2017a.
He, C., Takano, Y., Liou, K.-N., Yang, P., Li, Q., and Chen, F.: Impact of snow grain shape and black carbon-snow internal mixing on snow optical properties: Parameterizations for climate models. Journal of Climate, 30, 10,019–10,036, doi:10.1175/JCLI-D-17-0300.1, 2017b.
He, C., Liou, K.-N., Takano, Y., Yang, P., Qi, L., and Chen, F.: Impact of grain shape and multiple black carbon internal mixing on snow albedo: Parameterization and radiative effect analysis. J. Geophys. Res.-Atmos., 123, 1253–1268, doi:10.1002/2017JD027752, 2018a.
He, C., Liou, K.-N., and Takano, Y.: Resolving size distribution of black carbon internally mixed with snow: Impact on snow optical properties and albedo. Geophysical Research Letters, 45, 2697–2705, doi:10.1002/2018GL077062, 2018b.
Lee, W.-L., K. N. Liou, C. He, H.-C. Liang, T.-C. Wang, Q. Li, Z. Liu, and Q. Yue: Impact of absorbing aerosol deposition on snow albedo reduction over the southern Tibetan plateau based on satellite observations, Theor. Appl. Climatol., 129(3-4), 1373-1382, doi:10.1007/s00704-016-1860-4, 2017.
Li X., S. Kang, G. Zhang, B. Que, L. Tripatheea, R. Paudyal, Z. Jing, Y. Zhang, F. Yan, G. Li, X. Cui, R. Xu, Z. Hu, C. Li. Light-absorbing impurities in a southern Tibetan Plateau glacier: Variations and

potential impact on snow albedo and radiative forcing. Atmospheric Research, 200, 77-87, doi:10.1016/j.atmosres.2017.10.002, 2018.

Liou, K. N., Y. Takano, C. He, P. Yang, R. L. Leung, Y. Gu, and W. L. Lee: Stochastic parameterization for light absorption by internally mixed BC/dust in snow grains for application to climate models, J. Geophys. Res.-Atmos., 119, 7616–7632, doi:10.1002/2014JD021665, 2014.

Schwarz, J. P., Gao, R. S., Perring, A. E., Spackman, J. R., & Fahey, D. W. (2013). Black carbon aerosol size in snow. Scientific Reports, 3(1), 1356.

Zhang, Y., Kang, S., Sprenger, M., Cong, Z., Gao, T., Li, C., Tao, S., Li, X., Zhong, X., Xu, M., Meng, W., Neupane, B., Qin, X., and Sillanpää, M.: Black carbon and mineral dust in snow cover on the Tibetan Plateau, The Cryosphere, 12, 413-431, doi:10.5194/tc-12-413-2018, 2018.

**3.** Page 3, Line 9: Please remove "by" before "to explain".

**4.** Page 4, Lines 2-4: A recent study (He et al., 2018c) has updated a number of new features into the SNICAR model, including the effects of snow grain shape and aerosol-snow mixing state based on a set of new parameterizations (He et al., 2017b), which showed important impacts on aerosol-in-snow forcing. It seems that the authors here assumed external mixing between aerosols and spherical snow grains, which may not represent the realistic snowpack situation. It would be better if the authors could add some discussions on this important issue.
References:

He, C., Flanner, M. G., Chen, F., Barlage, M., Liou, K.-N., Kang, S., Ming, J., and Qian, Y.: Black carbon-induced snow albedo reduction over the Tibetan Plateau: Uncertainties from snow grain shape and aerosol-snow mixing state based on an updated SNICAR model, Atmos. Chem. Phys. Discuss., doi:10.5194/acp-2018-476, in review, 2018c.

**5.** Page 4, Lines 6-7: The authors focused on dust over the Tibetan Plateau by using a model spatial resolution of ~1 degree. However, this resolution may not be able to resolve the complex topography of the Tibetan Plateau and may cause some uncertainty in the simulations. Could the authors add some discussions on this aspect?

**6.** Page 4, Lines 12-13: The authors neglected the radiative properties of other aerosols, which may cause some biases in estimating dust-in-snow forcing. For example, Flanner et al. (2009) suggested that co-existing BC and dust may lead to smaller albedo reduction/forcing caused by dust (or BC) compared with dust (or BC)-only situation. Could the authors elaborate a little on this?

**7.** Page 4, Line 24: Please change "is" to "are".

**8.** Page 4, Lines 27-28: Could the small wet deposition of dust be due to the weak solubility of dust?

**9.** Section 2.2: (1) In terms of dust AOD, the authors only showed model results but no evaluation against observations, which seems not consistent with the section title "Model evaluation". It would be better if the authors could show some model evaluations on dust AOD (e.g., compare with satellite AOD during dust events). If this would take too much additional work, at least the authors could provide some references showing the evaluation of dust AOD using this model. (2) The authors showed some biases in modeled SCF, which may directly

translate into biases in dust-in-snow forcing. How would this bias affect the final results/conclusions? Could the authors add some discussions on this?

**10.** Section 3.1: The authors showed that the change in dust emissions induced by SRF+DRF is 5.98 Tg/season, which is contributed by two competing effects (-8.8 Tg/season caused by DRF and 14.78 Tg/season caused by SRF). It seems that the response of dust emissions to dust radiative effects is linear (5.98= -8.8+14.78), which may not be very intuitive, since some nonlinear processes (e.g., transport, deposition, circulation, etc.) are involved in this radiative feedback (Fig. 13). Could the authors add some comments on this?

**11.** Page 6, Lines 13-14: Another element in this positive feedback process is that increasing surface temperature leads to stronger snow aging and hence larger snow grain sizes, and finally reduces snow albedo.

**12.** Page 7, Lines 1-10: Could the authors put their SRF effects into the context? For example, are the results and conclusions shown here different from previous studies? If so, how different are they and why?

**13.** Page 8, Line 11: Another reason for the largest SRF in MAM could be that the snow cover/depth reaches the maximum over TP in early spring, along with the largest dust deposition, leading to the largest SRF.

**14.** Page 8, Line 21: It seems that the authors did not show results for the expansion of dust source region area caused by SRF in this manuscript.

---

## Referee Comment (RC2) · Anonymous Referee #2 · 18 Jun 2018

General Comments: The authors present a modeling study using the Community Atmosphere Model (CAM) with the bulk aerosol model (BAM) to investigate the positive radiative feedback of dust deposition on snow over East Asia. Due to the large scale climate impacts induced by Tibetan Plateau thermal forcing, this is a particularly important region to investigate. The study includes three 21-year simulations to isolate the radiative forcing associated with decreased snow cover and reflectivity due to dust deposition. There are some minor comments and questions that should be addressed along with deficiencies in overall grammar and sentence structure which are distracting to the content.

Specific Comments:

page 1, line 2: Please correct this sentence to, "...and removing snow cover through

increased snowmelt"

page 1, line 6-7: Please correct this sentence to, "Our results show that SRF increases the East Asian dust emissions by 13.7% in the spring, countering a 7.6% decrease in emissions by the . . ."

page 1, line 8: Please correct this to, ". . .dust cycle, including transport and deposition of dust aerosols over East Asia."

page 1, line 8-10: I suggest rephrasing this and removing "overall dust cycle over this region" as this has been stated in the previous sentence. "The simulations indicate an increase in dust emissions of 5.1% due to the combined effect of DRF and SRF."

page 1, line 12: Please correct this to, ". . .latent heat flux, which in turn increases the aridity and westerly winds over Northwest China and enhances the regional dust cycle."

page 2, line 3: is this eastern China or the east China Sea?

page 2, line 8: Please change "by" to "as"

page 2, line 13: Please rephrase this. The DRF with particle size distribution for dust from Kok, 2011, is less cooling (smaller forcing). The new size distribution results in DRF range of $\sim$-0.5—+0.2 W/m2 compared to AeroCom results of $\sim$ -0.8— - 0.01 W/m2 accounting for the possibility that the atmospheric dust burden could be a warming influence on net RF.

page 2,line 18: Please change "with" to "by"

page 2, line 23: Please change "depending" to "dependent"

page 2, line 31: Please remove "also" (last word on this line)

page 3, line 1-2: This is a run-on sentence. Please add a full stop after Qu et al., 2014 and start a new sentence with "These studies further claim. . ."

page 3, line 5: Please change to ". . .aerosol in snow can cause a $\sim$1 degree warming

of the TP, which. . ."

page 3, line 7: Replace "through" with "by"

page 3, line 9: Please remove "by" before "to explain"

page 3, line 13: Replace "affecting" with "impacts on"

page 3, line 19: Insert "been" between "not" and "studied"

page 3, line 19: Remove "here" and change "extended" to "extend"

page 3, line 20: Add a comma after SRF and remove "and" after SRF

page 3, lines 18-21: Run-on sentence, please break up this sentence either by adding a semicolon or making this two sentences.

page 3, line 23: Change "experiment design," to "experimental design." Remove "and furthermore" and begin new sentence "The model. . ."

page 3, line 24: Change "observations about the snow cover and the surface temperature" to "observations of snow cover and surface temperature." Change "The model results about" to "The model results for"

page 3, line 29: Remove "detailedly" and insert "in detail" between "described" and "by"

page 3, line 31: The improvements to CAM4-BAM are for the dust cycle only.

page 3, line 33-34 and p4, line 1: Can the authors be more specific about the impacts from the improved parameterizations? (e.g., dust in CAM4 release version was too absorbing and the size distribution was weighted too heavily on finest mode (bin1))

page 4, line 2: Insert "the" between "investigate" and "East", change "Asia" to "Asian" and replace "display its" with "the"

page 4, line 8-9: Can the authors add the sources of the input datasets used to drive the model? Please also describe the CAM4 and SNICAR setup. Please explain if these

are nudged simulations and if so, which meteorology data was used.

page 4: This study considers the dust-in-snow impact on RF but can the authors add some discussion on the combined deposition of all aerosols on snow? It would have been interesting to consider an additional case with deposition to snow from all radiatively active aerosols to shed light on the impact of dust-in-snow relative to, for example, BC+dust in snow.

page 4, line 16-17: Please rephrase the second part of this sentence to something like, "compared with changes induced solely by DRF"

Figure 2 (c) and (d): The color of the contour showing terrain > 2500m is difficult to see as it blends with the color scale in the figure.

page 4, line 24: Please rephrase, "...dust AOD has larger values over dust source regions (Gobi and Taklamakan deserts), where dust AOD is greater than 0.2 particularly over the Taklamakan desert."

page 4, line 25: Please insert "mean" between "monthly" and "dust"

page 4, line 26: Please combine "The result also indicates..." with the previous sentence, "...dry, wet and total (dry+wet) deposition, with the highest values in MAM."

page 4, line 27-28: I would rephrase "probably because of less rain over Northwest China." Consider adding citations for the trends in seasonal precipitation over this region (Xu et al., 2008; Kang et al., 2010; Fig. 8, Lau et al., 2006) Kang, S., Xu, Y., You, Q., Flügel, W.A., Pepin, N. and Yao, T., 2010. Review of climate and cryospheric change in the Tibetan Plateau.ÂăEnvironmental Research Letters,Âă5(1), p.015101. Xu, Z.X., Gong, T.L. and Li, J.Y., 2008. Decadal trend of climate in the Tibetan Plateau—regional temperature and precipitation.ÂăHydrological Processes,Âă22(16), pp.3056-3065.

page 4, line 29-31: Please change this sentence to, "Additionally, deserts in the western and northeastern regions of TP exhibit peaks in dust deposition, which we expect

to further increase the SRF signal."

page 5, Figure 3 (a) and (b): Can the authors provide further discussion on the low bias in the model for SCF during (MAM) and how this will impact their results? What are the implications of the low bias in surface temperature over the Gobi desert?

page 5, line 18: Replace "in the" before "MAM" with "during"

page 5, line 21: Typo "colume"

page 5, line 25: Change "remarkedly" to "markedly"

page 5, line 28: Are these statistically significant?

page 5, line 29: Please change "is" after "dust emissions" to "are"

page 5, line 30-31: Please insert "by 5.1%" between "East Asia" and "with"

page 6: The second paragraph in section 3.2 is good but can the authors comment on how well the simulated albedo in Figure 6 (a) matches satellite observations of albedo over this region?

page 6, line 17: Please change "That" after "Figure 7a)." to "This" and remove "that" between "because" and "the"

page 6: The negative RF from DRF over the Taklamakan and Gobi deserts from the scattering efficiency explained by simulated AOD would be more conclusive if the model AOD is compared to obs in the dust dominated regions.

page 6, line 20: Please change "decrease" after "surface" to "decreasing"

page 6, line 25: Please remove "reach" in "(reach above 20 W/m2)"

page 6, line 29: Please change "whereas" to "compared to"

page 6, line 30: Please change "determines" to "dominates"

page 6, line 33: Please change "followings." to "following sections."

page 7, line 8: Typo "(LHF+LHF)" (LHF+SHF)

page 7, line 9: Please change "resulting in" to "thereby"

page 7, line 12: Please add "referred to" between "TP" and "as" and change "monsoonal" to "monsoon"

page 7, line 19 and 20: I don't think "anonymous" is the right word. Do you mean anomalous?

page 7, line 21: Please change "That" between "regions." and "is" to "This" and remove "that" between "because" and "the"

page 7, line 28: Please remove "remarkedly" and add "the" between "affects" and "dust"

page 7, line 29: Please change to "which in turn increases the magnitude of the whole dust cycle"

page 7, line 34 and page 8, line 1: Please rephrase this sentence, in particular "decreased (increased)" and "over the north of the TP (over the TP)" is confusing.

page 8, line 3: Please change "generally" to "general" and "enhances" to "enhancing"

page 8, line 11: please change "that" to "the" between "because" and "largest"

page 8, line 14: Please rephrase to something like, "The above results indicate the predominate causes for SRF enhancement of the dust cycle over East Asia and is illustrated in Figure 13."

page 8, line 15: Please rephrase to something like, "Dust aerosols emitted from East Asian source regions where precipitation is limited and deposited . . ."

page 8, line 18: Please change to "enhancing the south-north temperature gradient and the aridity over. . ."

page 8, line 19: Please change "These" to "The"

page 8, line 23: Please change "we compares" to "we compare"

page 9, line 2-4: This sentence is grammatically incorrect, please try breaking it into two sentences or adding a semicolon.

page 9, line 7-9: Please change to "The CAM4-BAM simulations show that SRF increases dust emissions in the spring by 14.78Tg/season (13.7%), thus enhancing dust transport and deposition over East Asia."

page 9, line 12-14: Please change to something like, "Dust-in-snow reduces the albedo over the TP which warms the TP and enhances TP thermal effects and the regional dust cycle; increased sensible and latent heat fluxes from the surface result in increased aridity and westerly winds over North China."

page 9, line 14: Please change "In generally" to "In general"

page 9, line 15: Remove "overall"

---

## Referee Comment (RC3) · Anonymous Referee #3 · 25 Jun 2018

This is an interesting scientific paper in which the authors investigated the radiative feedbacks of dust in snow over East Asia by using CAM4 model simulations. The results are helpful for the scientists to understand the impact of dust-in snow on radiation balance and climate over East Asia. However, some details and figures should be supplemented and explained before published. (see specific comments).

Specific comments:
part of this study focus on TP, e.g. Fig.1~4, while other results are presented over East Asia, I suggest to present these results in a consistent way.
Sec.2.2. have you assessed the modeled AOD against satellite retrievals? The model results and conclusions in this study really depend on the modeled dust AOD. Additionally, absorption AOD (AAOD) also need to be assessed.
p.4, line.28. ' It is noted that the dry deposition of dusts is much larger than the wet deposition probably because of less rain over Northwest China'. It should be straightforward to present the comparison of precipitation to confirm this statement.
p.4 line.33. what is CRU? It should be explained before you cite it.
Sec.3.1 and Fig.5: how do you define the term trans in Fig.5?
p.5, line 30: 'Figures 5c, 5f, 5i and 5l show the changes in dust cycle induced by the dust total radiative forcing. The dust emissions are significantly enhanced (in Figure 5c) by the dust total radiative forcing over East Asia …', what are the physical mechanisms? It would be helpful to provide the dust emission scheme in the model and explain in detail why the dust EF enhance the dust emission.
p.6, line 12-15:'The decrease in snow albedo mainly results from a positive feedback process: absorbing aerosols deposited on snow - reducing surface albedo - increasing surface net solar radiation - increasing surface temperature -reducing snow fraction and depth- finally reducing surface albedo…'. I would suggest to present the physical variables listed above to support your conclusions. For example, surface net solar radiation, etc.
p.6, line 20: the authors mentioned that dust emissions are influenced by PBL mechanism, but never show that how PBL changes and how it modify the dust emission.
p.7, line 2: above -15%, change to 15% since you have stated it is decrease.
p.7, line 3: '…, and then expands the dust source region area…', where can you see the expanding?
Line 14: this is NOT recently, this is actually over 5 years ago.
Line 18: please define Omega before use it.
p.8 and Fig.12, please also give the total dust emission and the percentage of dust emissions induced by SRF to the total emission.

---

## Author Comment (AC1) · 13 Aug 2018

Response to Reviewer #1:

General comments:

The authors systematically investigated the responses of dust emissions, transport, and deposition to dust direct and in-snow radiative effects over East Asia. This work could help to improve the understanding of dust radiative effects and feedbacks in this region. The manuscript is generally well written and particularly I like Figure 13 which concisely summarizes the possible dust-in-snow radiative feedback. I have a few comments for improving the manuscript. Although most of my comments are minor, they need to be addressed properly before the manuscript can be considered for publication.

Response: Thank the Reviewer very much for the positive comments.

**1.** Page 2, Lines 11-13: As the authors mentioned, Kok et al. (2017) showed that inaccurate dust size distribution could lead to nontrivial biases in modeled DRF. Is it accurate enough by using the Bulk Aerosol Model (BAM) scheme embedded in CAM to represent dust size distributions as done in the present study?

Yes, we used the improved CAM4-BAM model as described by Albani et al. (2014) and Xie et al. (2018) in the present study. This CAM4-BAM model has been improved in terms of three major aspects: (1) optimized soil erodibility maps through generating the specific scale factors to the macroareas, (2) updated dust optical properties based on more realistic absorption coefficients (Yoshioka et al., 2007), and (3) an improved size distribution for use in dust emissions provided by Kok (2011). It is noted that the accurate dust size distribution (Kok et al., 2017) is from the analytical results (Kok et al., 2011), which is absolutely same with the improved CAM4-BAM used in our study.

**2.** Page 2, Lines 24-34: A number of recent references on advancing the understanding of BC/dust-in-snow effects are missing here. For example, several studies (e.g., Flanner et al., 2012; Liou et al., 2014; Dang et al., 2016; He et al., 2017b, 2018a) have shown the significant impacts of snow grain shape (spherical vs.

nonspherical) and aerosol-snow mixing state (internal vs. external) on BC/dust-snow albedo forcing. Further studies also investigated the effects of snow grain packing (e.g., He et al., 2017a) and aerosol size distribution in snow (e.g., Schwarz et al., 2013; He et al., 2018b) on aerosol-snow interactions. Since the aerosol-in-snow effect is the focus of this study, I suggest including these recent references here. In addition, in terms of BC/dust deposition over the TP (Lines 32-34), some latest observational studies (e.g., Lee et al., 2017; Li et al., 2018; Zhang et al., 2018) can also be included here.

References:

Dang, C., Q. Fu, and S. Warren: Effect of Snow Grain Shape on Snow Albedo, J. Atmos. Sci., 73, 3573–3583, doi: 10.1175/JAS-D-15-0276.1, 2016.

Flanner, M. G., X. Liu, C. Zhou, J. E. Penner, and C. Jiao: Enhanced solar energy absorption by internally-mixed black carbon in snow grains, Atmos. Chem. Phys., 12(10), 4699–4721, doi:10.5194/acp-12-4699-2012, 2012.

He, C., Y. Takano, and K.-N. Liou: Close packing effects on clean and dirty snow albedo and associated climatic implications, Geophys. Res. Lett., 44, doi:10.1002/2017GL072916, 2017a.

He, C., Takano, Y., Liou, K.-N., Yang, P., Li, Q., and Chen, F.: Impact of snow grain shape and black carbon-snow internal mixing on snow optical properties: Parameterizations for climate models. Journal of Climate, 30, 10,019–10,036, doi:10.1175/JCLI-D-17-0300.1, 2017b.

He, C., Liou, K.-N., Takano, Y., Yang, P., Qi, L., and Chen, F.: Impact of grain shape and multiple black carbon internal mixing on snow albedo: Parameterization and radiative effect analysis. J. Geophys. Res.-Atmos., 123, 1253–1268, doi:10.1002/2017JD027752, 2018a.

He, C., Liou, K.-N., and Takano, Y.: Resolving size distribution of black carbon internally mixed with snow: Impact on snow optical properties and albedo. Geophysical Research Letters, 45, 2697–2705, doi:10.1002/2018GL077062, 2018b.

Lee, W.-L., K. N. Liou, C. He, H.-C. Liang, T.-C. Wang, Q. Li, Z. Liu, and Q. Yue: Impact of absorbing aerosol deposition on snow albedo reduction over the

southern Tibetan plateau based on satellite observations, Theor. Appl. Climatol., 129(3-4), 1373-1382, doi:10.1007/s00704-016-1860-4, 2017.

Li X., S. Kang, G. Zhang, B. Que, L. Tripatheea, R. Paudyal, Z. Jing, Y. Zhang, F. Yan, G. Li, X. Cui, R. Xu, Z. Hu, C. Li. Light-absorbing impurities in a southern Tibetan Plateau glacier: Variations and potential impact on snow albedo and radiative forcing. Atmospheric Research, 200, 77-87, doi:10.1016/j.atmosres.2017.10.002, 2018.

Liou, K. N., Y. Takano, C. He, P. Yang, R. L. Leung, Y. Gu, and W. L. Lee: Stochastic parameterization for light absorption by internally mixed BC/dust in snow grains for application to climate models, J. Geophys. Res.-Atmos., 119, 7616–7632, doi:10.1002/2014JD021665, 2014.

Schwarz, J. P., Gao, R. S., Perring, A. E., Spackman, J. R., & Fahey, D. W. (2013). Black carbon aerosol size in snow. Scientific Reports, 3(1), 1356.

Zhang, Y., Kang, S., Sprenger, M., Cong, Z., Gao, T., Li, C., Tao, S., Li, X., Zhong, X., Xu, M., Meng, W., Neupane, B., Qin, X., and Sillanpää, M.: Black carbon and mineral dust in snow cover on the Tibetan Plateau, The Cryosphere, 12, 413-431, doi:10.5194/tc-12-413-2018, 2018.

Thank the Reviewer's comments for providing a number of recent references on advancing the understanding of BC/dust-in-snow effects. In the revised manuscript, we have added these references and the corresponding descriptions according to the comments: *"Recent studies have shown the significant impacts of snow grain shape (spherical vs. nonspherical) and aerosol-snow mixing state (internal vs. external) on BC/dust-in-snow radiative forcing (e.g., Flanner et al., 2012; Liou et al., 2014; Dang et al., 2016; He et al., 2017b, 2018a). Further studies were also conducted to investigate the effects of snow grain packing (He et al., 2017a) and aerosol size distribution in snow (Schwarz et al., 2013; He et al., 2018b) on aerosol-snow interactions."* And we also added *"There exists a larger amount of deposition on snow of black carbon and dust aerosols over the TP due to the high industrial and natural emissions in Asia from observational studies (Xu et al., 2009; Ming et al., 2013; Qu et al., 2014; Lee et al., 2017; Li et al., 2018; Zhang et al., 2018)."*

**3. Page 3, Line 9: Please remove "by" before "to explain".**

Taken.

**4.** Page 4, Lines 2-4: A recent study (He et al., 2018c) has updated a number of new features into the SNICAR model, including the effects of snow grain shape and aerosol-snow mixing state based on a set of new parameterizations (He et al., 2017b), which showed important impacts on aerosol-in-snow forcing. It seems that the authors here assumed external mixing between aerosols and spherical snow grains, which may not represent the realistic snowpack situation. It would be better if the authors could add some discussions on this important issue.

References:

He, C., Flanner, M. G., Chen, F., Barlage, M., Liou, K.-N., Kang, S., Ming, J., and Qian, Y.: Black carbon-induced snow albedo reduction over the Tibetan Plateau: Uncertainties from snow grain shape and aerosol-snow mixing state based on an updated SNICAR model, Atmos. Chem. Phys. Discuss., doi:10.5194/acp-2018-476, in review, 2018c.

We have added these references and the corresponding descriptions according to the comments: "Note that a set of new parameterizations including the effects of snow grain shape and aerosol-snow mixing state has been coupled into the SNICAR model, which may represent the realistic snowpack situation (He et al., 2018c). It is interesting to check the difference in radiative forcing over East Asia between these two models in the future. "

**5.** Page 4, Lines 6-7: The authors focused on dust over the Tibetan Plateau by using a model spatial resolution of ~1 degree. However, this resolution may not be able to resolve the complex topography of the Tibetan Plateau and may cause some uncertainty in the simulations. Could the authors add some discussions on this aspect?

Yes,we have added some discussions in the revised manuscript: "Due to the complex topography of the TP, higher-resolution simulations can resolve more details of the

deep valleys and high mountains over and around the TP and make some significant improvements in the simulated climate (Li et al., 2015). Hence, it is necessary to conduct the higher-resolution simulations to address this issue.*"*

**6.** Page 4, Lines 12-13: The authors neglected the radiative properties of other aerosols, which may cause some biases in estimating dust-in-snow forcing. For example, Flanner et al. (2009) suggested that co-existing BC and dust may lead to smaller albedo reduction/forcing caused by dust (or BC) compared with dust (or BC)-only situation. Could the authors elaborate a little on this?

Yes, I believe the Reviewer's point is exactly right. Generally, (A+B) effect = A effect + B effect + AB nonlinear interactions in the model. Hence, in the revised manuscript, we have added "It is noted that black carbon (BC) deposited on snow over the TP mainly from South Asia and East Asia (Xu et al., 2009; Wang et al., 2015) also displays a significant positive forcing over this region (Flanner et al., 2009; Qian et al., 2011). Here, we only consider the radiative forcing of the dust-in-snow over the TP ignoring the radiative forcing of the BC-in-snow in our study. Due to neglecting the nonlinear interactions between BC and dust, the dust-in-snow radiative forcing might not be accurate. Additionally, the overestimated SCF in the MAM may also artificially increase the dust-in-snow radiative forcing. The overestimated radiative forcing may amplify its feedbacks on the East Asian climate and dust cycle."

**7. Page 4, Line 24: Please change "is" to "are".**

Taken.

**8.** Page 4, Lines 27-28: Could the small wet deposition of dust be due to the weak solubility of dust?

Figure S1 shows the percentage of the dust wet deposition to the total deposition in the MAM. Over Northwest China, it has the smallest percentage of dust wet deposition and it has larger percentage of dust wet deposition over the Ocean. Hence, we can conclude that the less rain determines the small wet deposition over Northwest China based on the spatial distribution of the percentage.

[Figure]

Figure S1, Percentage of dust wet deposition to the total deposition (wet+dry deposition) in the MAM.

**9.** Section 2.2: (1) In terms of dust AOD, the authors only showed model results but no evaluation against observations, which seems not consistent with the section title "Model evaluation". It would be better if the authors could show some model evaluations on dust AOD (e.g., compare with satellite AOD during dust events). If this would take too much additional work, at least the authors could provide some references showing the evaluation of dust AOD using this model. (2) The authors showed some biases in modeled SCF, which may directly translate into biases in dust-in-snow forcing. How would this bias affect the final results/conclusions? Could the authors add some discussions on this?

Yes, I have added the descriptions about comparisons with observed AOD and deposition and the corresponding references (Albani et al., 2014; Xie et al., 2018). In the revised manuscript, we have added "This improved CAM4-BAM has been evaluated against measurements such as AOD, and dust deposition over the East Asia (Albani et al., 2014; Xie et al., 2018), showing a better simulation of dust cycle." We also added the overestimated SCF: "Additionally, the overestimated SCF in the MAM may also artificially increase the dust-in-snow radiative forcing. The overestimated radiative forcing may amplify its feedbacks on the East Asian climate and dust cycle."

**10.** Section 3.1: The authors showed that the change in dust emissions induced by SRF+DRF is 5.98 Tg/season, which is contributed by two competing effects (-8.8 Tg/season caused by DRF and 14.78 Tg/season caused by SRF). It seems that the response of dust emissions to dust radiative effects is linear (5.98= -8.8+14.78), which may not be very intuitive, since some nonlinear processes (e.g., transport, deposition, circulation, etc.) are involved in this radiative feedback (Fig. 13). Could the authors add some comments on this?

Yes, the Reviewer's point is exactly right. The total change in dust emissions induced by SRF+DRF is 5.98 Tg/season (Case1-Case3), which is resulted from the two competing effects. However, the changes caused by DRF (-8.8 Tg/season) and SRF (14.78 Tg/season) are included the nonlinear interactions between SRF and DRF. Hence, the values of dust emissions caused by DRF and SRF can be altered when removing the nonlinear interactions between SRF and DRF. Hence, in the revised manuscript, we have added "It is noted that the total change in dust emissions induced by SRF+DRF is 5.98 Tg/season, which is absolutely exact. However, the changes caused by DRF (-8.8 Tg/season) and SRF (14.78 Tg/season) are included the nonlinear interactions between SRF and DRF. Hence, the values of dust emissions caused by DRF and SRF can be altered when removing the nonlinear interactions between SRF and DRF."

**11.** Page 6, Lines 13-14: Another element in this positive feedback process is that increasing surface temperature leads to stronger snow aging and hence larger snow grain sizes, and finally reduces snow albedo.

Yes, we have added this element in the positive feedback process in the revised manuscript "Another element in this positive feedback process is that increasing surface temperature results in stronger snow aging and hence larger snow effective grain sizes, and finally reduces snow albedo (Flanner et al., 2009)."

**12.** Page 7, Lines 1-10: Could the authors put their SRF effects into the context? For example, are the results and conclusions shown here different from previous studies?

If so, how different are they and why?

Yes, we have added the descriptions "It is noted that SRF significantly increases the surface temperature, reduces the SCF and enhances the surface total heat flux (LHF and SHF) over the TP, which is absolutely same as the previous results (Qian et al., 2011). Due to the higher horizontal resolution of ~1 degree in this study, our result shows the finer spatial distribution of changes in these properties, especially over the TP compared to Qian et al. (2011)."

**13.** Page 8, Line 11: Another reason for the largest SRF in MAM could be that the snow cover/depth reaches the maximum over TP in early spring, along with the largest dust deposition, leading to the largest SRF.

Yes, we have added "This is mainly because the larger snow cover in MAM, along with the largest dust deposition exerts a significant radiative forcing, climatic feedbacks, and changes in dust emissions in this season."

**14.** Page 8, Line 21: It seems that the authors did not show results for the expansion of dust source region area caused by SRF in this manuscript.

As we know, dust emissions are primarily a function of surface wind speed, vegetation (and snow) cover, and soil erodibility. The decreases in vegetation and snow cover in the modeled grids can enhance the dust emissions by expanding the dust source area of the corresponding grids. Additionally, Figure S2 also shows that the total dust source area in our simulations is also expanded, due to the decreased snow cover by SRF. Hence, SRF can results in the expansion of dust source region area by reducing snow cover evidently.

[Figure]

Figure S2, Dust source area defined as emission flux>0 kg/m2/s with Case 1 (Real line) and Case 2 (dotted line).

---

## Author Comment (AC2) · 13 Aug 2018

Response to Reviewer #2:

General comments:

General Comments: The authors present a modeling study using the Community Atmosphere Model (CAM) with the bulk aerosol model (BAM) to investigate the positive radiative feedback of dust deposition on snow over East Asia. Due to the large scale climate impacts induced by Tibetan Plateau thermal forcing, this is a particularly important region to investigate. The study includes three 21-year simulations to isolate the radiative forcing associated with decreased snow cover and reflectivity due to dust deposition. There are some minor comments and questions that should be addressed along with deficiencies in overall grammar and sentence structure which are distracting to the content.

Response: Thank the Reviewer very much for the positive comments. We have tried our best to enhance the English in grammar and sentence structure.

Specific Comments:

page 1, line 2: Please correct this sentence to, ": : :and removing snow cover through increased snowmelt"

Taken.

page 1, line 6-7: Please correct this sentence to, "Our results show that SRF increases the East Asian dust emissions by 13.7% in the spring, countering a 7.6% decrease in emissions by the : : :"

Taken.

page 1, line 8: Please correct this to, ": : :dust cycle, including transport and deposition of dust aerosols over East Asia."

Taken.

page 1, line 8-10: I suggest rephrasing this and removing "overall dust cycle over this region" as this has been stated in the previous sentence. "The simulations indicate an

increase in dust emissions of 5.1% due to the combined effect of DRF and SRF."

Taken.

page 1, line 12: Please correct this to, ": : :latent heat flux, which in turn increases the aridity and westerly winds over Northwest China and enhances the regional dust cycle."

Taken.

page 2, line 3: is this eastern China or the east China Sea?

Here, we claimed that dust aerosols can be carried over the wide downwind regions including land (e.g., the eastern China) and sea (e.g., the Pacific ocean).

page 2, line 8: Please change "by" to "as"

Taken.

page 2, line 13: Please rephrase this. The DRF with particle size distribution for dust from Kok, 2011, is less cooling (smaller forcing). The new size distribution results in DRF range of _-0.5âˇAˇ T+0.2 W/m2 compared to AeroCom results of _ -0.8âˇAˇT -0.01 W/m2 accounting for the possibility that the atmospheric dust burden could be a warming influence on net RF.

We have rewritten it as "The DRF with particle size distribution for dust from Kok (2011) is less cooling (smaller forcing) because atmospheric dust is coarser than represented in current models. The new size distribution results in DRF range of -0.48 W m$^{-2}$ and +0.2 W m$^{-2}$, including the possibility that dust causes a net warming of the planet (Kok et al., 2017)."

page 2,line 18: Please change "with" to "by"

Taken.

page 2, line 23: Please change "depending" to "dependent"

Taken.

page 2, line 31: Please remove "also" (last word on this line)

Taken.

page 3, line 1-2: This is a run-on sentence. Please add a full stop after Qu et al., 2014 and start a new sentence with "These studies further claim: : :"

Taken.

page 3, line 5: Please change to ": : :aerosol in snow can cause a _1 degree warming of the TP, which: : :"

Taken.

page 3, line 7: Replace "through" with "by"

Taken.

page 3, line 9: Please remove "by" before "to explain"

Taken.

page 3, line 13: Replace "affecting" with "impacts on"

Taken.

page 3, line 19: Insert "been" between "not" and "studied"

Taken.

page 3, line 19: Remove "here" and change "extended" to "extend"

Taken.

page 3, line 20: Add a comma after SRF and remove "and" after SRF

Taken.

page 3, lines 18-21: Run-on sentence, please break up this sentence either by adding a semicolon or making this two sentences.

Taken.

page 3, line 23: Change "experiment design," to "experimental design." Remove "and furthermore" and begin new sentence "The model: : :"

Taken.

page 3, line 24: Change "observations about the snow cover and the surface temperature" to "observations of snow cover and surface temperature." Change "The model results about" to "The model results for"

Taken.

page 3, line 29: Remove "detailedly" and insert "in detail" between "described" and "by"

Taken.

page 3, line 31: The improvements to CAM4-BAM are for the dust cycle only.

Taken.

page 3, line 33-34 and p4, line 1: Can the authors be more specific about the impacts from the improved parameterizations? (e.g., dust in CAM4 release version was too absorbing and the size distribution was weighted too heavily on finest mode (bin1))

Yes, we have added the descriptions in the revised manuscript "This improved size distribution decreases the emitted fraction of clay aerosols (< 2 um) in excellent agreement with measurements and exerts a smaller cooling compared released version. "

page 4, line 2: Insert "the" between "investigate" and "East", change "Asia" to "Asian" and replace "display its" with "the"

Taken.

page 4, line 8-9: Can the authors add the sources of the input datasets used to drive the model? Please also describe the CAM4 and SNICAR setup. Please explain if these are nudged simulations and if so, which meteorology data was used.

Yes, it is a good suggestion. We have added the input datasets "The SST and sea-ice concentration were from a merged version of the HadISST (Rayner et al., 2003) and the optimum interpolation SST data sets described by Hurrell et al. (2008). We conducted three numerical experiments including 21-year free run with a 1-year spin up (no nudging), one with both DRF and SRF (Case1), one with the DRF and without the SRF (Case2), the other one without the DRF and the SRF (Case3), as summarized in Table 1."

page 4: This study considers the dust-in-snow impact on RF but can the authors add some discussion on the combined deposition of all aerosols on snow? It would have been interesting to consider an additional case with deposition to snow from all radiatively active aerosols to shed light on the impact of dust-in-snow relative to, for example, BC+dust in snow.

Yes, we have added a discussion about BC+dust in snow "It is noted that black carbon (BC) deposited on snow over the TP mainly from South Asia and East Asia (Xu et al., 2009; Wang et al., 2015) also displays a significant positive forcing over this region (Flanner et al., 2009; Qian et al., 2011). Here, we only consider the radiative forcing of the dust-in-snow over the TP ignoring the radiative forcing of the BC-in-snow in our study. Due to neglecting the nonlinear interactions between BC and dust, the dust-in-snow radiative forcing might not be accurate."

page 4, line 16-17: Please rephrase the second part of this sentence to something like, "compared with changes induced solely by DRF"

Taken.

Figure 2 (c) and (d): The color of the contour showing terrain > 2500m is difficult to see as it blends with the color scale in the figure.

Taken.

page 4, line 24: Please rephrase, ": : :dust AOD has larger values over dust source regions (Gobi and Taklamakan deserts), where dust AOD is greater than 0.2 particularly over the Taklamakan desert."

Taken.

page 4, line 25: Please insert "mean" between "monthly" and "dust"

Taken.

page 4, line 26: Please combine "The result also indicates: : :" with the previous sentence, ": : :dry, wet and total (dry+wet) deposition, with the highest values in MAM."

Taken. It is a very good suggestion. Thanks very much.

page 4, line 27-28: I would rephrase "probably because of less rain over Northwest China." Consider adding citations for the trends in seasonal precipitation over this region (Xu et al., 2008; Kang et al., 2010; Fig. 8, Lau et al., 2006) Kang, S., Xu, Y., You, Q., Flügel, W.A., Pepin, N. and Yao, T., 2010. Review of climate and cryospheric change in the Tibetan Plateau.¢aEnvironmental Research Letters, ¢ a5(1), p.015101. Xu, Z.X., Gong, T.L. and Li, J.Y., 2008. Decadal trend of climate in the Tibetan Plateauâ˘A ˘Tregional temperature and precipitation. aHydrological Processes,¢a22(16), pp.3056-3065.

Taken.

page 4, line 29-31: Please change this sentence to, "Additionally, deserts in the western and northeastern regions of TP exhibit peaks in dust deposition, which we expect to further increase the SRF signal."

Taken.

page 5, Figure 3 (a) and (b): Can the authors provide further discussion on the low bias in the model for SCF during (MAM) and how this will impact their results? What are the implications of the low bias in surface temperature over the Gobi desert?

The bias in the model for SCF and surface temperature over the TP region are mainly due to the model's coarser horizontal resolution ($0.9^{o}*1.25^{o}$). Hence, we have added the corresponding descriptions in the revised manuscript "Due to the complex topography of the Tibetan Plateau, higher-resolution simulations can resolve more details of the deep valleys and high mountains over and around the TP and make some significant improvements in the simulated climate (Li et al., 2015). Hence, it is necessary to conduct the higher-resolution simulations to address this issue. " And we also added "Due to neglecting BC and dust nonlinear interactions, the dust-in-snow radiative forcing might not be accurate. Additionally, the overestimated SCF in the MAM may also artificially increase the dust-in-snow radiative forcing. The overestimated radiative forcing may amplify its feedbacks on the East Asian climate and dust cycle."

page 5, line 18: Replace "in the" before "MAM" with "during"

Taken.

page 5, line 21: Typo "colume"

Taken.

page 5, line 25: Change "remarkedly" to "markedly"

Taken.

page 5, line 28: Are these statistically significant?

We have deleted the word "significantly". The changes in dust transport and dry deposition are statistically significant whereas the change in wet deposition in not statistically significant.

page 5, line 29: Please change "is" after "dust emissions" to "are"

Taken.

page 5, line 30-31: Please insert "by 5.1%" between "East Asia" and "with"

Taken.

page 6: The second paragraph in section 3.2 is good but can the authors comment on how well the simulated albedo in Figure 6 (a) matches satellite observations of albedo over this region?

Yes, we have added a description in the revised manuscript "Compared to the MODIS surface albedo over the TP (Meng et al., 2018), the CAM4-BAM model captures its spatial distribution during MAM. However, the model overestimates the surface albedo, which is similar with multi-model ensembles' results (Li et al., 2016), mainly due to the overestimated SCF and the ignoring BC-in-snow."

page 6, line 17: Please change "That" after "Figure 7a)." to "This" and remove "that" between "because" and "the"

Taken.

page 6: The negative RF from DRF over the Taklamakan and Gobi deserts from the scattering efficiency explained by simulated AOD would be more conclusive if the model AOD is compared to obs in the dust dominated regions.

Taken.

page 6, line 20: Please change "decrease" after "surface" to "decreasing"

Taken.

page 6, line 25: Please remove "reach" in "(reach above 20 W/m2)"

Taken.

page 6, line 29: Please change "whereas" to "compared to"

Taken.

page 6, line 30: Please change "determines" to "dominates"

Taken.

page 6, line 33: Please change "followings." to "following sections."

Taken.

page 7, line 8: Typo "(LHF+LHF)" (LHF+SHF)

Taken.

page 7, line 9: Please change "resulting in" to "thereby"

Taken.

page 7, line 12: Please add "referred to" between "TP" and "as" and change "monsoonal" to "monsoon"

Taken.

page 7, line 19 and 20: I don't think "anonymous" is the right word. Do you mean anomalous?

Taken.

page 7, line 21: Please change "That" between "regions." and "is" to "This" and remove "that" between "because" and "the"

Taken.

page 7, line 28: Please remove "remarkedly" and add "the" between "affects" and "dust"
Taken.

page 7, line 29: Please change to "which in turn increases the magnitude of the whole dust cycle"
Taken.

page 7, line 34 and page 8, line 1: Please rephrase this sentence, in particular "decreased (increased)" and "over the north of the TP (over the TP)" is confusing.
Yes, It has been changed as "It shows the downward vertical velocity is enhanced in summer (Figure 11a), resulting in the significant decreased surface precipitation (Figure 11b) over the north of the TP, whereas the upward vertical velocity and the surface precipitation are both enhanced over the TP."

page 8, line 3: Please change "generally" to "general" and "enhances" to "enhancing"
Taken.

page 8, line 11: please change "that" to "the" between "because" and "largest"
Taken.

page 8, line 14: Please rephrase to something like, "The above results indicate the predominate causes for SRF enhancement of the dust cycle over East Asia and is illustrated in Figure 13."
Taken.

page 8, line 15: Please rephrase to something like, "Dust aerosols emitted from East Asian source regions where precipitation is limited and deposited : : :"

Taken.

page 8, line 18: Please change to "enhancing the south-north temperature gradient and the aridity over: : :"
Taken.

page 8, line 19: Please change "These" to "The"
Taken.

page 8, line 23: Please change "we compares" to "we compare"
Taken.

page 9, line 2-4: This sentence is grammatically incorrect, please try breaking it into two sentences or adding a semicolon.
Taken.

page 9, line 7-9: Please change to "The CAM4-BAM simulations show that SRF increases dust emissions in the spring by 14.78Tg/season (13.7%), thus enhancing dust transport and deposition over East Asia."
Taken.

page 9, line 12-14: Please change to something like, "Dust-in-snow reduces the albedo over the TP which warms the TP and enhances TP thermal effects and the regional dust cycle; increased sensible and latent heat fluxes from the surface result in increased aridity and westerly winds over North China."
Taken.

page 9, line 14: Please change "In generally" to "In general"
Taken.
page 9, line 15: Remove "overall"

Taken.

---

## Author Comment (AC3) · 13 Aug 2018

Response to Reviewer #3:

General comments:

This is an interesting scientific paper in which the authors investigated the radiative feedbacks of dust in snow over East Asia by using CAM4 model simulations. The results are helpful for the scientists to understand the impact of dust-in snow on radiation balance and climate over East Asia. However, some details and figures should be supplemented and explained before published. (see specific comments).
Response: Thank the Reviewer very much for the positive comments.

Specific comments:
part of this study focus on TP, e.g. Fig.1~4, while other results are presented over East Asia, I suggest to present these results in a consistent way.
Thank the Reviewer very much. Because Figures 1-4 are mainly used to compare the climate (including surface temperature, snow cover fraction and so on) and dust properties over the TP, these figures are focused on the TP. However, dust-in-snow over the TP can affect the climate and dust cycle over a much larger region (e.g., the whole East Asia). Hence, the other figures include a larger region to order to investigate its climate effects.

Sec.2.2. have you assessed the modeled AOD against satellite retrievals? The model results and conclusions in this study really depend on the modeled dust AOD. Additionally, absorption AOD (AAOD) also need to be assessed.
Thanks. In the Reference (Xie et al., 2018), the improved CAM4-BAM was used to evaluate the East Asian dust, including surface dust concentrations and the AOD, based on local observational sites from CAWNET (Zhang et al., 2012) and CARSNET (Che et al., 2015). Additionally, scatter and absorption of dusts exerts a radiative forcing for surface, TOA, and Atmosphere, which has also been discussed detailedly in the Reference (Xie et al., 2018). In this manuscript, we focus on comparing the surface temperature and snow cover fraction in order to investigate

dust-in-snow forcing.

Reference

Che, H., Zhang, X. Y., Xia, X., Goloub, P., Holben, B., Zhao, H., ⋯ Blarel, L. (2015). Ground-based aerosol climatology of China: Aerosol optical depths from the China Aerosol Remote Sensing Network (CARSNET) 2002–2013. Atmospheric Chemistry and Physics, 15(8), 7619–7652.

Xie, X., Liu, X., Che, H., Xie, X., Wang, H., Li, J., ⋯ Liu, Y. (2018). Modeling East Asian dust and its radiative feedbacks in CAM4-BAM. Journal of Geophysical Research: Atmospheres, 123, 1079–1096. https://doi.org/10.1002/2017JD027343

Zhang, X. Y., Wang, Y. Q., Niu, T., Zhang, X. C., Gong, S. L., Zhang, Y. M., & Sun, J. Y. (2012). Atmospheric aerosol compositions in China: Spatial/temporal variability, chemical signature, regional haze distribution and comparisons with global aerosols. Atmospheric Chemistry and Physics, 12, 779–799.

p.4, line.28. ' It is noted that the dry deposition of dusts is much larger than the wet deposition probably because of less rain over Northwest China'. It should be straightforward to present the comparison of precipitation to confirm this statement.

Response: Figure S1 shows the percentage of the dust wet deposition to the total deposition in the MAM. Over Northwest China, it has the smallest percentage of dust wet deposition and it has larger percentage of dust wet deposition over the Ocean. Hence, we can conclude that the less rain determines the small wet deposition over Northwest China based on the spatial distribution of the percentage.

[Figure]

Figure S1, Percentage of dust wet deposition to the total deposition (wet+dry deposition) in the MAM.

p.4 line.33. what is CRU? It should be explained before you cite it.

Taken.

Sec.3.1 and Fig.5: how do you define the term trans in Fig.5?

Response: The dust transport $Q$ is the vertically integrated dust flux, which is similar to the water vapor transport,

$$Qu = \frac{1}{g} \int_{ps}^{100} au\,\mathrm{d}p$$

$$Qv = \frac{1}{g} \int_{ps}^{100} av\,\mathrm{d}p$$

$$Q = (Qu^2 + Qv^2)^{1/2}$$

where a, u, v, and g represent dust mass concentration, zonal wind, meridional wind, pressure and gravitational acceleration, respectively. ps is the surface pressure. Hence, in our manuscript, we have added the corresponding descriptions *"defined as the vertically integrated dust flux, which is similar to water vapor transport."*

p.5, line 30: 'Figures 5c, 5f, 5i and 5l show the changes in dust cycle induced by the dust total radiative forcing. The dust emissions are significantly enhanced (in Figure 5c) by the dust total radiative forcing over East Asia …', what are the physical mechanisms? It would be helpful to provide the dust emission scheme in the model

and explain in detail why the dust EF enhance the dust emission.

Yes, It is a very good suggestion. In the previous paper (Xie et al., 2018), we have shown the total vertical dust flux $F_d$ (unit: kg m$^{-2}$ s$^{-1}$) from the soil during saltation is calculated as

$$F_d = C_{MB}\eta f_{bare}\frac{\rho}{g}u_*^3\left(1 - \frac{u_{*t}^2}{u_*^2}\right)\left(1 + \frac{u_{*t}}{u_*}\right), (u_* \geq u_{*t}), \tag{1}$$

$$F_d = 0, (u_* < u_{*t}), \tag{2}$$

More atmospheric instability and larger 10 m wind speed can both enhance the dust emission flux by increasing u$_*$. Our further analysis reveals that these results are mainly due to the regional climatic feedbacks induced by SRF over East Asia. By reducing the snow albedo over the TP, the dust-in-snow mainly warms the TP to enhance its thermal effects by increasing the surface sensible and latent heat flux, and then increases the aridity and westerly winds over Northwest China, in turn enhances the East Asian dust cycle.

p.6, line 12-15:'The decrease in snow albedo mainly results from a positive feedback process: absorbing aerosols deposited on snow -reducing surface albedo -increasing surface net solar radiation -increasing surface temperature -reducing snow fraction and depth-finally reducing surface albedo…'. I would suggest to present the physical variables listed above to support your conclusions. For example, surface net solar radiation, etc.

Thanks very much for the Reviewer. The physical variables are shown in the corresponding figures. The increasing surface forcing has been shown in the Figure 7c, the increasing surface temperature has been presented in Figure 7d, and the reducing snow cover has been shown in Figure 8a.

p.6, line 20: the authors mentioned that dust emissions are influenced by PBL mechanism, but never show that how PBL changes and how it modify the dust emission.

Response: It is good suggestion. The dust direct radiaitive forcing (DRF) reduces the dust emissions by the PBL mechanism, which has been detailedly described in the previous paper. In this manuscript, we focus on investigating the dust-in-snow radiative forcing (SRF) and its feedbacks on the regional climate and the dust cycle over East Asia, only compared with DRF's results.

p.7, line 2: above -15%, change to 15% since you have stated it is decrease.
Taken.

p.7, line 3: '…, and then expands the dust source region area…', where can you see the expanding?
As we know, dust emissions are primarily a function of surface wind speed, vegetation (and snow) cover, and soil erodibility. The decreases in vegetation and snow cover in the modeled grids can enhance the dust emissions by expanding the dust source area of the corresponding grids. Additionally, Figure 2 also shows that the total dust source area in our simulations is also expanded, due to the decreased snow cover by SRF. Hence, SRF can results in the expansion of dust source region area by reducing snow cover evidently.

[Figure]

Figure S2, Dust source area defined as emission flux>0 kg/m2/s with Case 1 (Real line) and Case 2 (dotted line).

Line 14: this is NOT recently, this is actually over 5 years ago.
Taken.

Line 18: please define Omega before use it.
Taken.

p.8 and Fig.12, please also give the total dust emission and the percentage of dust emissions induced by SRF to the total emission.

Response: It is noted that Table 2 shows the dust emissions for DRF, SRF and DRF+SRF. The change in dust emissions induced by DRF is -8.80 Tg season$^{-1}$, whereas the change induced by SRF is +14.78 Tg season$^{-1}$, the total change by DRF+SRF is 5.98 Tg season$^{-1}$ in Table 2. The sign of changes induced by DRF and SRF is absolutely opposite. Hence, the percentage (247%) of dust emissions induced by SRF to the total emission is meaningless.

**Table 2.** The March-April-May (MAM) averaged dust emissions (Tg season$^{-1}$), transport (g m$^{-1}$ s$^{-1}$), dry deposition (Tg season$^{-1}$), and wet deposition (Tg season$^{-1}$) over the East Asian dust source area (75°E−115°E and 25°N−50°N) in Case1, Case2, and Case3, as well as their corresponding differences between these three experiments.

|  | Dust emission | Dust transport | Dry deposition | Wet deposition |
|---|---|---|---|---|
| Case1 | 122.40 | 1.08 | 68.92 | 36.99 |
| Case2 | 107.62 | 1.01 | 61.59 | 35.33 |
| Case3 | 116.42 | 1.09 | 65.33 | 35.96 |
| DRF (Case2−Case3) | −8.80 (−7.6%) | −0.07 (−6.4%) | −3.74 (−5.7%) | −0.63 (−1.8%) |
| SRF (Case1−Case2) | 14.78 (13.7%) | 0.07 (6.9%) | 7.33 (11.9%) | 1.66 (4.7%) |
| DRF+SRF (Case1−Case3) | 5.98 (5.1%) | −0.01 (−0.9%) | 3.59 (5.5%) | 1.03 (2.9%) |